# Changes in working and health conditions in different sectors between 2010 and 2019 in a representative sample of French workers

Jean-Francois Gehanno[1,2]*, Ariane Leroyer[3,4], Manon Couvreur[5], Serge Volkoff[6], Laetitia Rollin[1,2]

**1** Institute of Occupational Medicine, Rouen University Hospital, Rouen, France, **2** Inserm, Rouen University, Sorbonne University, University of Paris 13, Laboratory of Medical Informatics and Knowledge Engineering in e-Health, LIMICS, Paris, France, **3** Lille University Hospital, Public Health, Epidemiology, and Economic Health, Lille, France, **4** Lille University, Inserm, U1286-INFINITE-Institute for Translational Research in inflammation, Lille, France, **5** Observatoire régional de la santé et du social, Rouen, France, **6** Centre d'études de l'emploi et du travail, Creapt, Saint-Denis, France

☯ These authors contributed equally to this work.
* jf.gehanno@chu-rouen.fr

## Abstract

### Background

Studies on evolution of working conditions and workers' health in developed countries show conflicting results, and often fail to identify evolutions by occupational sector. This study aimed to describe recent evolutions of working conditions and health status of French workers over a decade across different activity sectors.

### Methods and findings

Data from a large national French observatory of working conditions and health were used. This observatory collects, for a random sample of employees, data on their working conditions and health. Data on working conditions and health status of employees in the 13 main French activity sectors in 2010–2011 and 2018–2019 were used. A 2 steps weighting method was used to allow results to be extrapolated to the French employees.

Overall, 23,951 and 26,538 questionnaires were collected for the 2010–2011 and 2018–2019 periods, respectively. Job demand significantly decreased and job resources significantly increased between 2010–2011 and 2018–2019 while physical workload decreased. Neck, wrist, elbow and back pain disorders significantly decreased as well as anxiety, whereas shoulder disorders, fatigue and sleep disorders increased. However, there were large and significant disparities in evolutions between sectors, for exposures and health status. The most worrisome sector was the Human health and social work sector with no improvement in the working conditions and increasing health problems.

**Data availability statement:** Data cannot be shared publicly due to legal reasons related to data protection and confidentiality. The data of the EVREST French observatory can be accessed on the basis of a research project submitted to the executive committee. The contact point for accessing the data is the Data Protection Officer of the University of Rouen: dpo@univ-rouen.fr More information can be found on the following website: https://evrest.istnf.fr/_docs/Fichier/2017/4-170512013717.pdf

**Funding:** The author(s) received no specific funding for this work.

**Competing interests:** The authors have declared that no competing interests exist.

## Conclusion

Overall, evolutions and prevalence of physical exposure or health problems do not bring the same information. It is important to understand the extent to which psychosocial, physical working conditions and health status differ among various industry categories, and to study the determinants of such disparities such that policy and practice interventions to reduce disparities can be appropriately directed. It is also important to continue monitoring health of workers, as the impact of changes in working conditions may not have an immediate effect, particularly for conditions with a long latency period.

## Introduction

Globalization of market, job automation and digitalization, work extensity and physical hazards are some of the most important drivers of workers' health changes [1]. However, it is difficult to get a reliable picture of the evolution of working conditions, if any, and there is uncertainty whether they are changing for the better or worse.

In Europe, estimates derived from the cross-national European Working Conditions Survey [2], suggested job insecurity, skill discretion and decision latitude had deteriorated over the 2005–2010 period while other work conditions, such as working hours and work-life imbalance, had improved. Comparing data from two surveys of employees in European Union member-states carried out in 1996 and 2001, Gallie found no evidence of a trend towards higher work pressure over this period [3], and a study exploring the development of working conditions within and between occupations in the Swedish labour market from 1997 to 2015 found that the global analysis of the work environment of occupations revealed few changes over time [4]. In the US, analyzing the NIOSH Quality of Work Life Surveys, Myers et al. observed statistically significant increases in job strain, low job control and work-family conflict [5] whereas, in Canada, Fan and Smith found improvements in some self-reported measures of the psychosocial work environment between 2002 and 2012 [6].

Studying global trends is important to identify new risk factors, assess if occupational health interventions on the national level are successful, and quantify the health impact of working conditions. However, it neglects possible differences between occupations [7]. The Swedish study pointed out that, although macro trends remained the same, the working conditions within different occupations were changing and most of the dimensions displayed occupational variation in trajectories over time [4].

Studying the global trends in workers' health is subject to the same bias. Over the last decades, the nature of health problems has changed with emerging concerns around musculoskeletal disorders and mental health [8], but more precise data on the trends in different occupations are needed.

This study aimed to describe recent evolutions of working conditions (*e.g.,* employee perceptions of psychosocial factors or physical exposures) and health status (musculoskeletal and psychological issues) of workers followed by occupational medicine services over a decade across different activity sectors using data from a large national French observatory of working conditions and health.

## Method

Data from the National Evrest Observatory (EVolutions et Relations en Santé au Travail − Evolutions and Relationships in Health at Work) were used.

In France, all employees are required to attend an occupational health examination every two, three or five years, depending on the employee's level of risk exposure (http://evrest.istnf.fr/page-0-0-0.html). Evrest is a national observatory, set up in France in 2007 by occupational health physicians and researchers aiming at collecting, for a random sample of employees, a database concerning their working conditions and their health. The scope of this observatory covers all employees in the public and private sectors working in mainland France, excluding agriculture and the state civil service, which includes ministries and national public administrative establishments. Since 2008, the Evrest database has included about 180,000 questionnaires, completed by about 80,000 employees and 1,700 occupational health teams.

Data are collected using a short, standardized questionnaire (one double-sided page), made up of closed-ended questions that have been used in large surveys on health in workplaces in France [9–12] (http://evrest.istnf.fr/page-0-0-0.html).

Working conditions are studied using data about psychosocial and biomechanical constraints.

The psychosocial constraints are explored according to 5 main axes as recommended by the College d'expertise sur le suivi statistique des risques psychosociaux au travail (College of experts on the statistical monitoring of psychosocial risks at work) [13], *i.e.,* job demand (intensity and working time), margin of maneuver and autonomy, social support and reward, ethical dilemmas and job insecurity. The biomechanical constraints are explored according to 3 items: awkward postures, effort or carrying loads and repetitive movements.

Psychological distress is estimated based on the presence of one of the following symptoms: fatigue, sleeping disorders or anxiety-nervousness and on the presence of all these 3 symptoms during the last 7 days. Musculoskeletal complaints of the upper limbs are estimated based on the presence of shoulder, elbow or wrist/hand complaints. Musculoskeletal complaints of the spine are estimated based on the presence of dorsal/lumber vertebrae or cervical vertebrae complaints (Table 1).

In the questionnaire, sectors of activity and occupations are coded according to the national classifications of the National Institute of Statistics and Economic Studies, PCS-ESE-2003 and NAF-2008, respectively (https://www.insee.fr/en/metadonnees/nafr2/section/A).

The questionnaire is proposed by occupational health teams (physicians and nurses) participating in the survey to employees during their periodic occupational health examination. All other types of examination (resuming work after sick leave, spontaneous visit for health complaints, pre-employment visit) are excluded.

The part of the questionnaire concerning working conditions is completed by the employee in the waiting room and the second part, concerning state of health, is completed by the occupational health physician or the nurse during the visit.

To obtain a random sample of employees, only those born in October are asked to complete the questionnaire.

## Population studied

Workers surveyed as part of the Evrest program in 2010–2011 and in 2018–2019 were included.

However, we included in the study only sectors of activity for which we had at least 500 questionnaires, which led us to include 13 main activity sectors.

- C: Manufacturing

- D: Electricity, gas, steam and air conditioning supply

- F: Constructions and construction works

- G: Wholesale and retail trade; repair of motor vehicles and motorcycles

- H: Transportation and storage services

**Table 1. Dimensions description and scales used.**

| Domain and dimension | Dimension formulation | Response scale in present study | Question number |
|---|---|---|---|
| **Job demands** | | | |
| Time pressure | Due to your workload, do you ever skip or shorten meal times, not take a break? | 1 = never, 2 = rarely, 3 = quite often, 4 = very often | 1 |
| Time pressure | Due to your workload, do you ever complete too quickly an operation that should have required more attention? | 1 = never, 2 = rarely, 3 = quite often, 4 = very often | 2 |
| Overtime | Due to your workload, do you ever work overtime? | 1 = never, 2 = rarely, 3 = quite often, 4 = very often | 3 |
| Task interruptions | Are you frequently disturbed in your work by having to stop working on one task to complete another unforeseen one? | 1 = yes, 2 = no | 4 |
| Ethical dilemma | Do you have the means to do your work to a high standard? | 1 = not at all, 2 = Mostly no, 3 = mostly yes, 4 = yes, absolutely | 5 |
| Ethical dilemma | Do you have to do things you disapprove? | 1 = not at all, 2 = Mostly no, 3 = mostly yes, 4 = yes, absolutely | 6 |
| Job insecurity | Do you work with the fear of losing your job? | 1 = not at all, 2 = Mostly no, 3 = mostly yes, 4 = yes, absolutely | 7 |
| **Job resources** | | | |
| Autonomy | Can you choose how you do things yourself? | 1 = not at all, 2 = Mostly no, 3 = mostly yes, 4 = yes, absolutely | 8 |
| Social support | Do you have enough possibilities to obtain help and cooperation? | 1 = not at all, 2 = Mostly no, 3 = mostly yes, 4 = yes, absolutely | 9 |
| Reward | Is your work recognized by your professional entourage? | 1 = not at all, 2 = Mostly no, 3 = mostly yes, 4 = yes, absolutely | 10 |
| Learning opportunities | Does your work provide opportunities to learn new things? | 1 = not at all, 2 = Mostly no, 3 = mostly yes, 4 = yes, absolutely | 11 |
| **Physical workload** | | | |
| | Are you exposed to repetitive movements? | 0 = no, never, 1 = yes, sometimes, 2 = yes, often | 12 |
| | Are you exposed to awkward postures? | 0 = no, never, 1 = yes, sometimes, 2 = yes, often | 13 |
| | Are you exposed to carrying heavy loads? | 0 = no, never, 1 = yes, sometimes, 2 = yes, often | 14 |
| **Health status** | | | |
| Musculoskeletal disorders | Neck pain or symptoms | 1 = yes, 2 = no | 15 |
| | Back pain or symptoms | 1 = yes, 2 = no | 16 |
| | Wrist pain or symptoms | 1 = yes, 2 = no | 17 |
| | Elbow pain or symptoms | 1 = yes, 2 = no | 18 |
| | Shoulder pain or symptoms | 1 = yes, 2 = no | 19 |
| Psychological | Sleep disorders during the last week? | 1 = yes, 2 = no | 20 |
| | Fatigue during the last week? | 1 = yes, 2 = no | 21 |
| | Anxiety during the last week? | 1 = yes, 2 = no | 22 |

- I: Accommodation and food service activities

- J: Information and communication services

- K: Financial and insurance services

- M: Professional, scientific and technical services

- N: Administrative and support service activities

- O: Public administration and defense

- Q: Human health and social work activities

- S: Other services activities

**Weighting methodology**

A weighting methodology allowing results to be extrapolated to the French employees was used. The detailed weighted methodology is available in a previously published article [14]. Briefly, the weighting method includes two steps: 1) a first weighting based on the date of the last occupational health examination, to consider the probability of participation of each employee; and 2) a marginal calibration method to correct potential distortions of the sample in comparison with the scope of the survey, based on the distribution of French employees in 2015 (middle of the studied period) in terms of gender, age (5 groups), company size (6 groups) and sector of activity crossed with occupations (13x4 groups) (National Institute of Statistics and Economic Studies, https://www.insee.fr/fr/statistiques/3536754).

**Statistical analysis**

Data from workers surveyed as part of the Evrest program in 2010–2011 on the one hand, and in 2018–2019 on the other hand, were compared.

If necessary, each working condition indicator was dichotomized, by grouping together: for time pressure and overtime, the responses "quite often" and "very often" in one hand and "never" and "rarely" on the other; for ethical dilemma, job insecurity, job resources, autonomy, social support, reward and learning opportunities, the responses "quite often" and "very often" in one hand and "not at all" and "mostly no" on the other; for physical workload, the response "yes, often" was opposed to the responses "no never" and "yes, sometimes".

Decrease of exposure to job demand (i.e., from "quite often" or "very often" to "never" or "rarely"), decrease of exposure to physical workload (from "yes, often" to "yes, sometimes" or "no never"), decrease of frequency of musculoskeletal or psychological disorders (from "yes" to "no") or increase of job resources (from "not at all" or "Mostly no" to "mostly yes" or "yes, absolutely") was considered as an improvement, the opposite being considered as a deterioration.

For each working condition and health indicator studied, the proportion of workers who were concerned was calculated for each period, and a proportion difference test for weighted data (Rao-Scott) was performed to test the evolution of the situation between the two periods. The Rao-Scott method introduced a correction into the classic chi-square test in order to took into account the weights as well as, if any, the clustering and stratification involved in the collection of the sample[15].

The significance threshold was set at 5%, and analyses were performed with R version 4.1.2.

**Ethics statement**

The recruitment period for this study started on January 1, 2010, ended December 31, 2011 for the first period and on January 1, 2018, and ended December 31, 2019 for the second period.

Before participation, employees were informed of their rights in writing and they had the choice to decline completing the questionnaire and participating to the study. No minor was included in the study.

All personal data are anonymized in the database. The observatory and the database obtained authorization from the French Data Protection Authority (Commission nationale de l'informatique et des libertés, CNIL, Authorization number 906290V1) and the National Advisory Committee on the Processing of Health Research Information (Authorization number 06.390) in 2006. The observatory and the database comply with the European General Data Protection Regulation (EU 2016/679).

**Results**

Description of the population.

Overall, 23,951 and 26,538 questionnaires were collected for the 2010–2011 and 2018–2019 periods, respectively (Table 2).

Male to female ratio remained close between the two periods (from 1.42 to 1.46). The proportion of workers aged 25–34 increased by 1%, and the number of employees and manual workers decreased by 2.3%.

The global prevalence and trends of exposure to risk factors and health issues are presented in Fig 1.

Each radar displays the proportion of workers who were exposed to the dimension of job demand or of physical workload mentioned in Table 1, the proportion of workers who declared having one or several job resources mentioned in Table 1 and the proportion of workers who suffered of Musculoskeletal or psychological disorders

Overall, job demand and job resources significantly improved between 2010–2011 and 2018–2019 while physical workload decreased. Concerning health status, neck, wrist, elbow and back pain disorders significantly improved as well as anxiety, whereas shoulder disorders, fatigue and sleep disorders increased.

However, there were large disparities in trends between sectors for all domains and dimensions (exposure or health status) (Figs 2–14)

**Table 2. Description of the population included (weighted percentages).**

| Sector of activity | Years | N | Gender (%) | | Age (%) | | | | | Occupation (%) | | | |
|---|---|---|---|---|---|---|---|---|---|---|---|---|---|
| | | | Female | Male | < 25 | 25-34 | 35-44 | 45-54 | ≥55 | M | IP | E | MW |
| Manufacturing | 2010-2011 | 4018 | 42.5 | 57.5 | 12.7 | 22.1 | 26.1 | 24.7 | 14.4 | 5.6 | 9.6 | 39.7 | 45.1 |
| | 2018-2019 | 3346 | 40.0 | 60.0 | 17.6 | 18.3 | 24.0 | 24.2 | 15.9 | 5.6 | 9.6 | 39.7 | 45.1 |
| Electricity, gas, steam and air conditioning supply | 2010-2011 | 2305 | 43.0 | 57.0 | 7.2 | 23.3 | 24.5 | 37.2 | 7.9 | 4.6 | 7.0 | 79.1 | 9.2 |
| | 2018-2019 | 3761 | 40.0 | 60.0 | 6.7 | 30.1 | 28.2 | 24.7 | 10.2 | 4.8 | 7.4 | 76.2 | 11.5 |
| Constructions and construction works | 2010-2011 | 1641 | 31.6 | 68.4 | 18.9 | 22.2 | 25.9 | 19.1 | 13.9 | 0.9 | 3.5 | 40.4 | 55.3 |
| | 2018-2019 | 2230 | 35.2 | 64.8 | 13.5 | 24.2 | 24.0 | 24.1 | 14.2 | 0.9 | 3.5 | 40.4 | 55.3 |
| Wholesale and retail trade | 2010-2011 | 3909 | 45.0 | 55.0 | 16.9 | 28.0 | 24.9 | 19.2 | 11.0 | 6.9 | 46.3 | 37.4 | 9.3 |
| | 2018-2019 | 3718 | 44.2 | 55.8 | 19.3 | 27.3 | 22.6 | 20.1 | 10.7 | 7.4 | 46.9 | 36.0 | 9.8 |
| Transportation and storage services | 2010-2011 | 1528 | 20.8 | 79.2 | 6.0 | 22.6 | 27.3 | 31.3 | 12.9 | 5.8 | 41.6 | 16.5 | 36.1 |
| | 2018-2019 | 1467 | 26.0 | 74.0 | 6.6 | 21.9 | 25.2 | 28.8 | 17.5 | 6.5 | 42.2 | 15.7 | 35.6 |
| Accommodation and food service | 2010-2011 | 875 | 48.0 | 52.0 | 30.4 | 24.7 | 18.6 | 17.1 | 9.2 | 5.2 | 26.1 | 59.9 | 8.7 |
| | 2018-2019 | 1017 | 45.7 | 54.3 | 34.3 | 21.9 | 18.5 | 17.7 | 7.6 | 3.3 | 27.1 | 60.7 | 8.8 |
| Information and communication | 2010-2011 | 664 | 28.4 | 71.6 | 12.0 | 32.6 | 27.8 | 19.9 | 7.7 | 53.6 | 36.4 | 8.4 | 1.5 |
| | 2018-2019 | 903 | 27.7 | 72.3 | 15.3 | 28.4 | 22.3 | 20.9 | 13.1 | 43.9 | 42.9 | 12.5 | 0.6 |
| Finance and insurance | 2010-2011 | 988 | 55.3 | 44.7 | 9.6 | 24.6 | 21.8 | 24.2 | 19.8 | 23.8 | 42.5 | 33.5 | 0.2 |
| | 2018-2019 | 696 | 53.8 | 46.2 | 10.8 | 28.3 | 27.3 | 19.5 | 14.2 | 21.3 | 49.1 | 28.7 | 1.0 |
| Professional, scientific and technical | 2010-2011 | 1317 | 43.3 | 56.7 | 10.1 | 31.8 | 26.2 | 18.9 | 13.0 | 34.3 | 41.9 | 20.8 | 2.9 |
| | 2018-2019 | 1481 | 44.3 | 55.7 | 13.8 | 31.1 | 25.0 | 20.1 | 10.0 | 36.6 | 36.8 | 23.9 | 2.7 |
| Administrative and support service | 2010-2011 | 1484 | 42.8 | 57.2 | 18.1 | 26.2 | 26.2 | 17.3 | 12.1 | 8.4 | 25.0 | 41.5 | 25.1 |
| | 2018-2019 | 1848 | 41.4 | 58.6 | 19.8 | 25.3 | 22.2 | 21.3 | 11.3 | 12.4 | 25.7 | 40.9 | 20.9 |
| Public administration and defense | 2010-2011 | 1009 | 59.2 | 40.8 | 2.7 | 16.9 | 23.5 | 32.5 | 24.3 | 24.1 | 26.0 | 39.2 | 10.6 |
| | 2018-2019 | 749 | 52.5 | 47.5 | 4.6 | 14.8 | 24.4 | 32.9 | 23.2 | 14.4 | 27.9 | 45.8 | 11.9 |
| Human health and social work | 2010-2011 | 2428 | 69.8 | 30.2 | 8.5 | 22.1 | 22.8 | 27.2 | 19.4 | 6.6 | 59.2 | 18.6 | 15.6 |
| | 2018-2019 | 3436 | 71.3 | 28.7 | 8.5 | 24.7 | 24.1 | 24.6 | 18.1 | 6.6 | 59.2 | 18.6 | 15.6 |
| Other services activities | 2010-2011 | 719 | 58.2 | 41.8 | 15.1 | 25.7 | 22.5 | 21.8 | 14.8 | 34.4 | 43.2 | 19.3 | 3.0 |
| | 2018-2019 | 583 | 59.3 | 40.7 | 15.0 | 22.5 | 22.8 | 23.0 | 16.8 | 36.4 | 39.2 | 21.0 | 3.4 |
| Global | 2010-2011 | 23951 | 50.2 | 49.8 | 12.1 | 23.7 | 24.2 | 24.5 | 15.5 | 14.2 | 32.7 | 32.6 | 20.6 |
| | 2018-2019 | 26538 | 50.2 | 49.8 | 12.1 | 23.7 | 24.2 | 24.5 | 15.5 | 14.2 | 32.7 | 32.6 | 20.6 |

N: Number of individuals; M: Managers; IP: Intermediate professions; E: Employees; MW: Manual workers

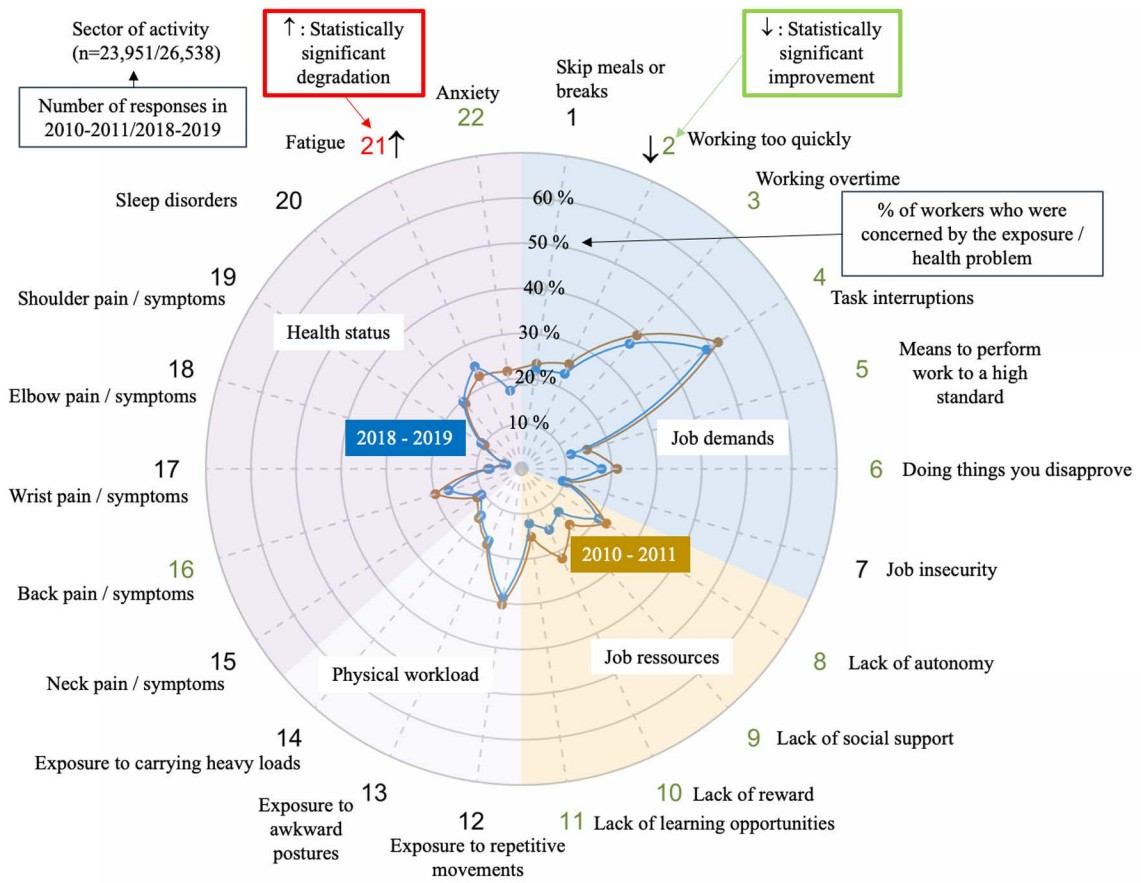

**Fig 1. Figure interpretation guide and global results.**

These differences were significantly different among all sectors (Table 3).

## Evolutions of working conditions

Social support, Reward and Learning opportunities improved for all sectors within the decade, but at different levels. Improvement was also globally observed for all the other psychosocial dimensions but with exceptions for some sectors. For example, the percentage of employees who feared losing their job was divided by two in the Information and communication services sector (from 13.2% to 6.8%) but nearly doubled for the Electricity, gas, steam and air conditioning supply and Public administration and defense sectors (from 3.8% to 7.4%).

Working overtime and being exposed to task interruptions were the two most prevalent psychosocial risk factors across all sectors of activity.

Physical workload remained unchanged within the period except for the task of Carrying heavy loads which significantly decreased in three sectors (Financial and insurance services, Professional, scientific and technical services and Administrative and support service activities).

Two sectors were particularly exposed to repetitive movements: Manufacturing and Accommodation and food service activities

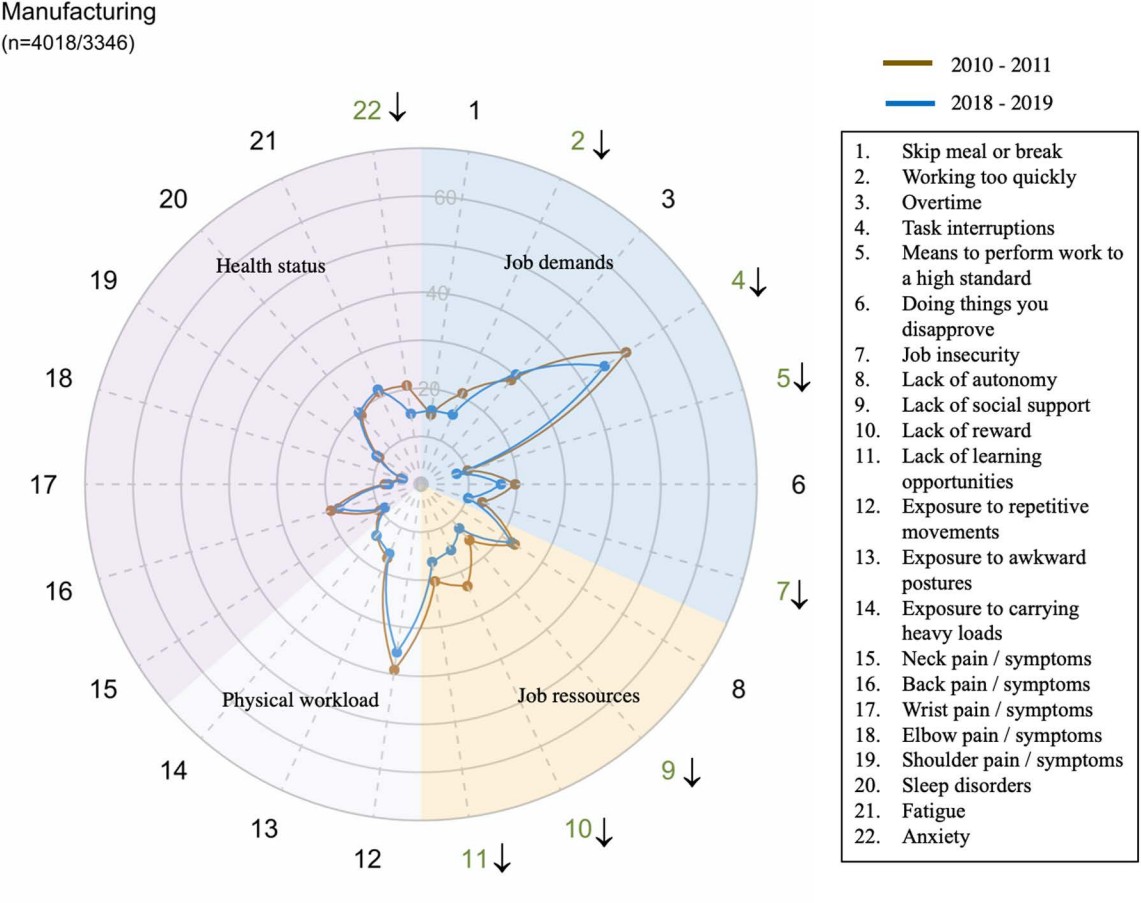

**Fig 2. Evolution of the percentage of workers exposed to psychosocial or physical risk factors and their health status in the manufacturing sector.**

## Evolutions of health status

For health status, the differences were even more pronounced, with some sectors showing improvement and others showing the opposite for the same dimension. The differences between sectors were significant for all dimensions.

Musculoskeletal disorders of the back, wrist and elbows decreased in many sectors whereas shoulder's disorders significantly increased in several sectors (*e.g.,* +75% for the shoulder in the Information and communication services sector).

Concerning psychological issues, the present study examined the development of fatigue and anxiety in 13 sectors of activity. The results revealed that these two phenomena have evolved in opposite ways in 10 of the sectors studied. This contrasting trend was found to be statistically significant in seven of these ten sectors.

## Relations between exposures and health problems

Trends in working conditions and health status were in general concordant within each sector.

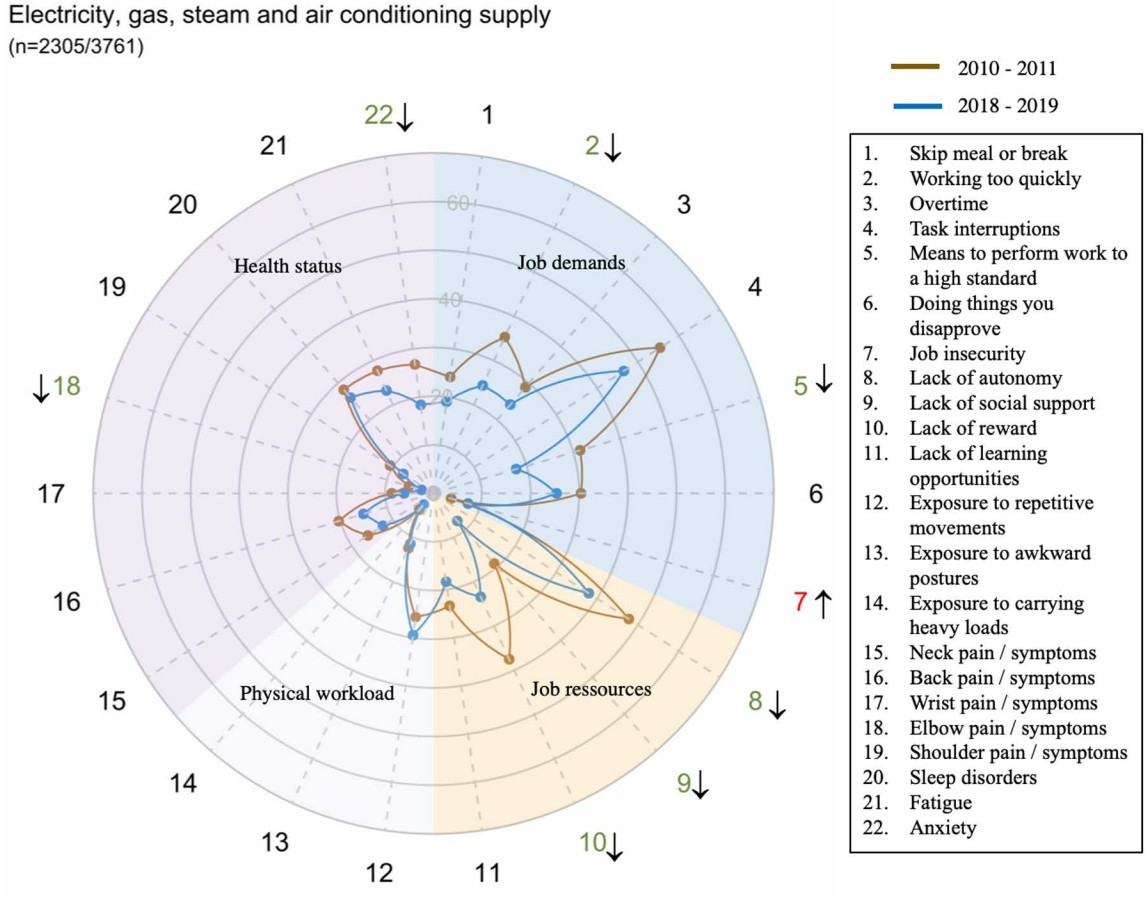

**Fig 3. Evolution of the percentage of workers exposed to psychosocial or physical risk factors and their health status in the electricity, gas, steam and air conditioning supply sector.**

For sectors C, F, H, M and S, working conditions and health status remained stable or improved over the period.

However, while working conditions remained stable or improved for the Human health and social work activities sector (Q), 7 out of 8 health disorders increased, significantly for 4 of them.

This was also the case, to a lesser extent, in the Wholesale and retail trade; repair of motor vehicles and motorcycles and the Information and communication services sector (G and J).

## Looking beyond trends

An analysis of the changes that have occurred between the two periods under study in the various sectors of activity may provide a misleading picture of the level of exposure to occupational risk factors or of the state of health of employees in these sectors.

In certain sectors, a substantial decline in exposure or health impairment has been observed; however, residual levels persist at elevated levels in comparison to other sectors. For example, task interruptions significantly decreased in the Information and communication services, the Financial and insurance services and the Professional, scientific and technical services sectors. However, they still concerned more than 55% of employees in 2018–2019. On the other side,

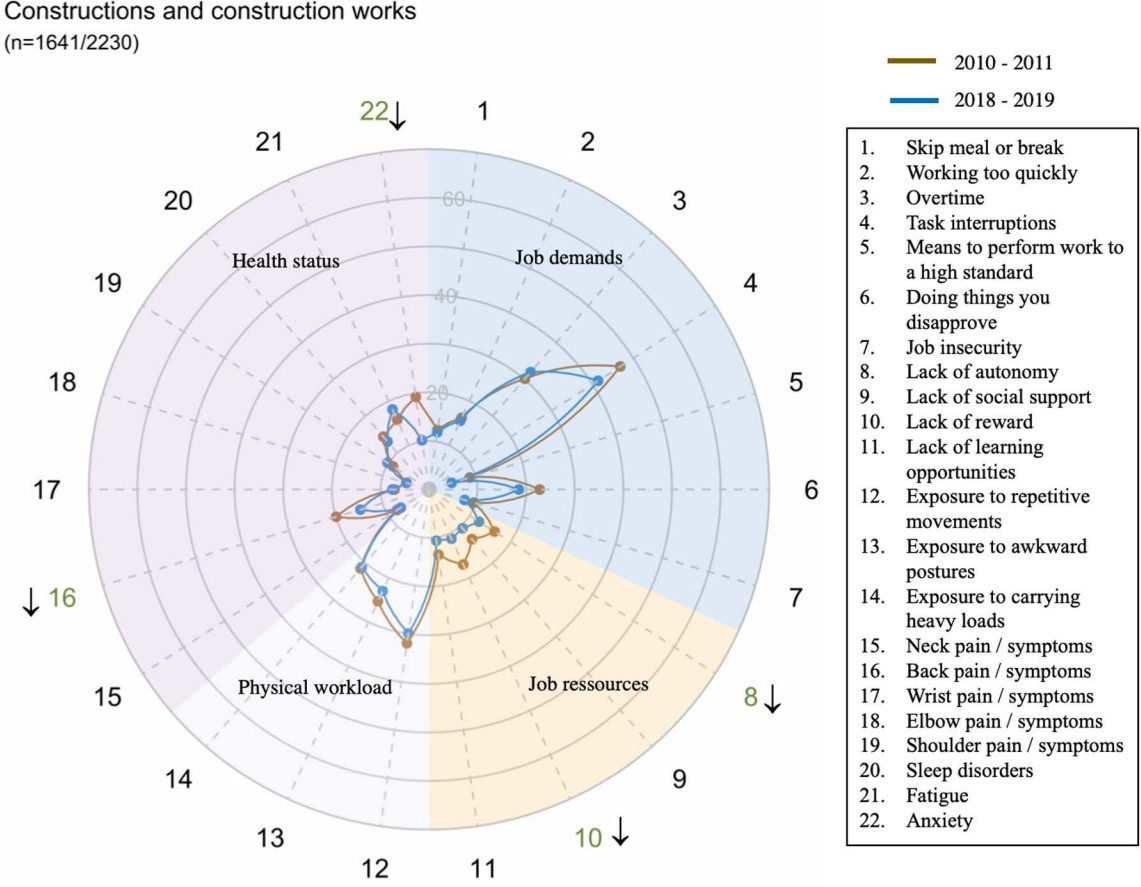

**Fig 4. Evolution of the percentage of workers exposed to psychosocial or physical risk factors and their health status in the construction sector.**

prevalence of task interruptions did not improve for the Transportation and storage services sector between the two periods but concerned "only" 45.7% of employees. While job insecurity significantly increased in the Financial and insurance services sector and significantly decreased in the Administrative and support service activities sector, it still concerned in 2018–2019 14.0% of employees in the Administrative and support service activities sector compared to 11.4% in the Financial and insurance services sector.

## Discussion

The main results of this study are a global improvement of health status, job demand and job resources while physical workload remained at the same level, among French workers between 2010–2011 and 2018–2019, with marked differences across sectors of activity.

## Limitations

A limitation of our study is that data were collected before COVID. The COVID-19 pandemic changed people's working conditions worldwide, induced income losses, increased work stressors and impaired the mental health of workers

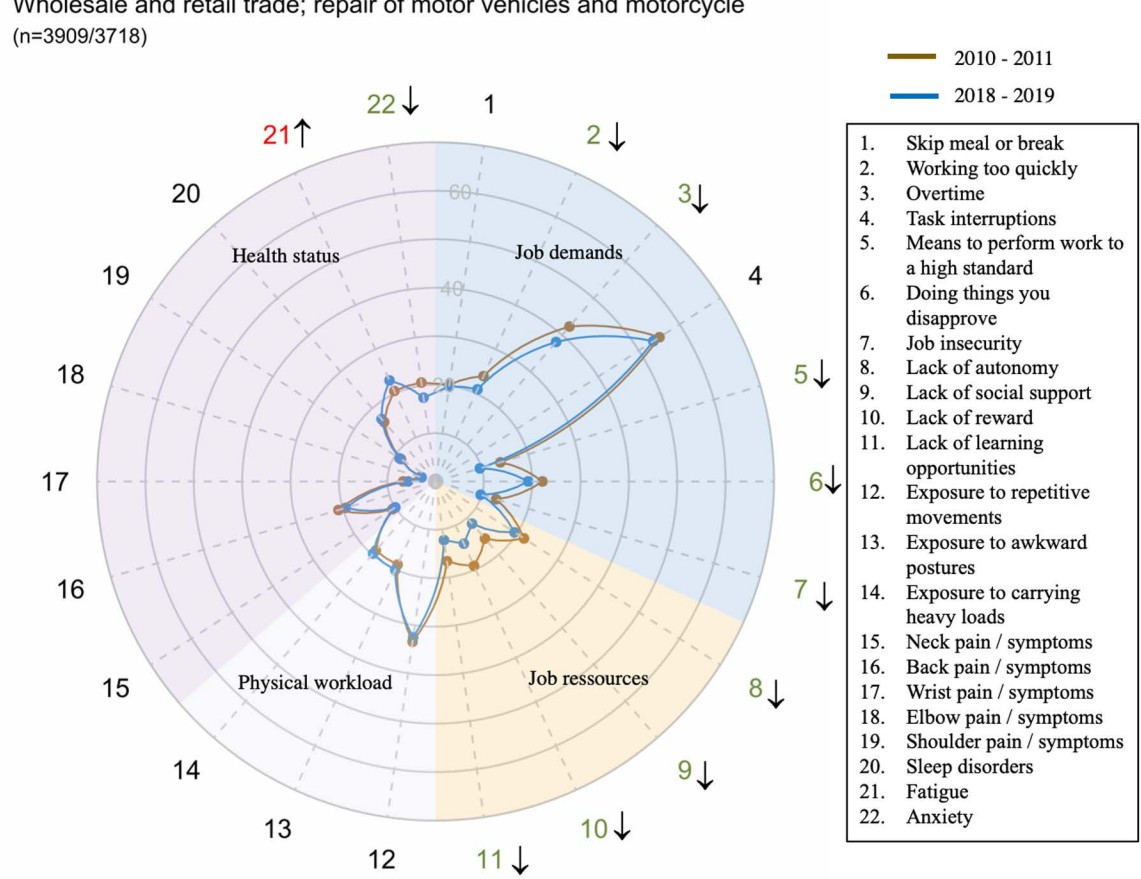

**Fig 5. Evolution of the percentage of workers exposed to psychosocial or physical risk factors and their health status in the wholesale and retail trade sector.**

[7,16,17]. However, most studies on trends in working conditions and health are in the same situation, due mainly to the delay in collecting and analysing data at a national level. Furthermore, knowing the trends before the pandemic will be important to analyze the changes it induced.

Another limitation is that this study is longitudinal, but does not follow the same individuals over time. Internationally, there are several major longitudinal cohort studies with a focus on work environment and health. Some of them, such as the Whitehall II study in England [18], the French GAZEL cohort [19] and the Finnish Public sector study [20] have multiple repeated measures on a range of factors concerning work, private life and health. All of these studies are, however, restricted to specific groups of employees: civil servants, employees at a gas or electricity company, or public sector employees, respectively. Our study, based on a representative sample, allowed us to compare trends between sectors of activity.

We studied 13 sectors of activity, among the 21 registered in the French National Institute of Statistics and Economic Studies database. We had not enough data for the 8 other sectors of activity, which employ a limited number of workers.

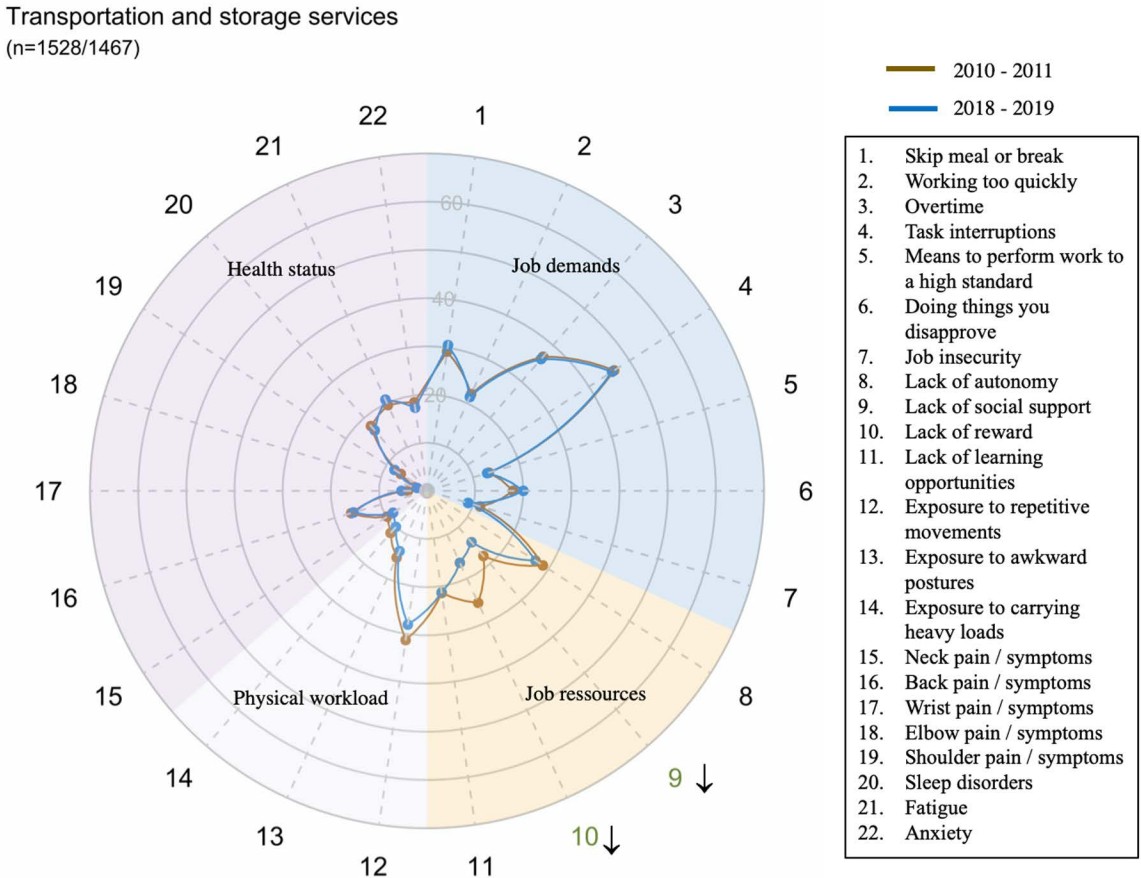

**Fig 6. Evolution of the percentage of workers exposed to psychosocial or physical risk factors and their health status in the transportation and storage services sector.**

We set the threshold at 500 questionnaires at least for one sector of activity and for both periods. For example, there were only 170, 47 and 20 workers included for "Arts, entertainment and recreation", "Mining and quarrying" and "Activities of households as employers" sectors, respectively. The agricultural sector (sector A 'Agriculture, forestry and fishing') was also very poorly represented (only 77 workers included 2010), as it is subject to specific occupational health monitoring by a special social security system. It was therefore excluded from the survey.

It should be noted that all employees in an industry category were classified according to their main activity. This implies that different job tasks that might affect job demands, control and social support or physical exposure, such as administrators, service workers, managers, or workers performing the core production, are merged and that the impact of these different job tasks could not be assessed. However, it is noteworthy that despite this limitation, several significant differences between sectors were found. To get more precise estimates, information on occupation could be used for stratification of workers performing the core tasks from other functions in the production units, for example.

The measures of the demands, decision authority and support dimensions that were used in the present study are not directly comparable with the standard measures of the demand-control-support model of Karasek and Theorell. However,

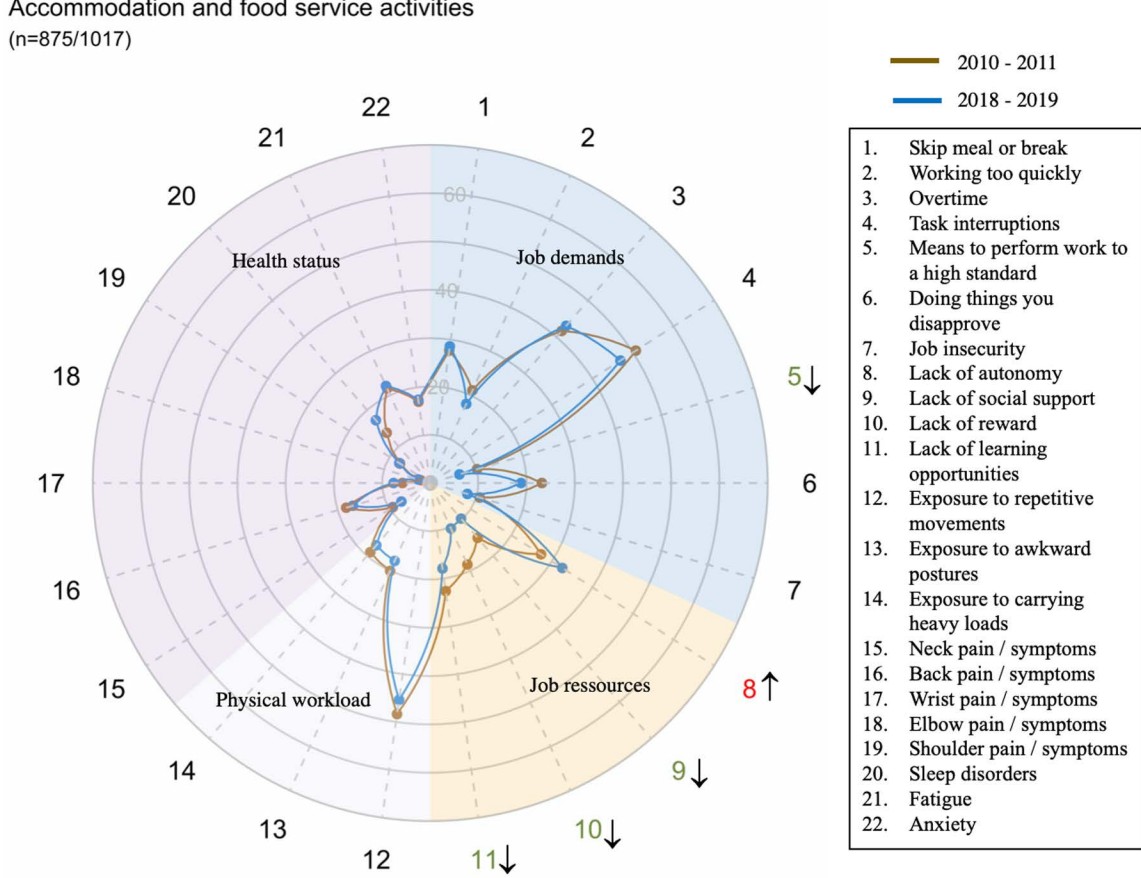

**Fig 7. Evolution of the percentage of workers exposed to psychosocial or physical risk factors and their health status in the accommodation and food service activities sector.**

these measures allow for comparisons between industry categories and over time and have been used in previous studies [12].

The way to dichotomize the scales of response for some working conditions, considering the responses "yes" in one hand and "no" in the other hand, does not take into account that for some of these working conditions, there may have been changes in frequency inside categories, such an increase of responses "very often" and a decrease of response "quite often" for time pressure, leading to a stability of the "yes" response.

Finally, our sample was restricted to those who were employees at the time of the survey and therefore self-employed or unemployed people were not included.

## Strenghts

The following strengths of our study should be underlined.

We measured both work environment and health extensively. Yet, our study was based on a large nationally representative sample of the French working population of employees. The number of employees included with more than 20,000 per period, as compared to the 9,000 initially included in the Swedish Longitudinal Occupational Survey of Health [21].

Information and communication services
(n=664/903)

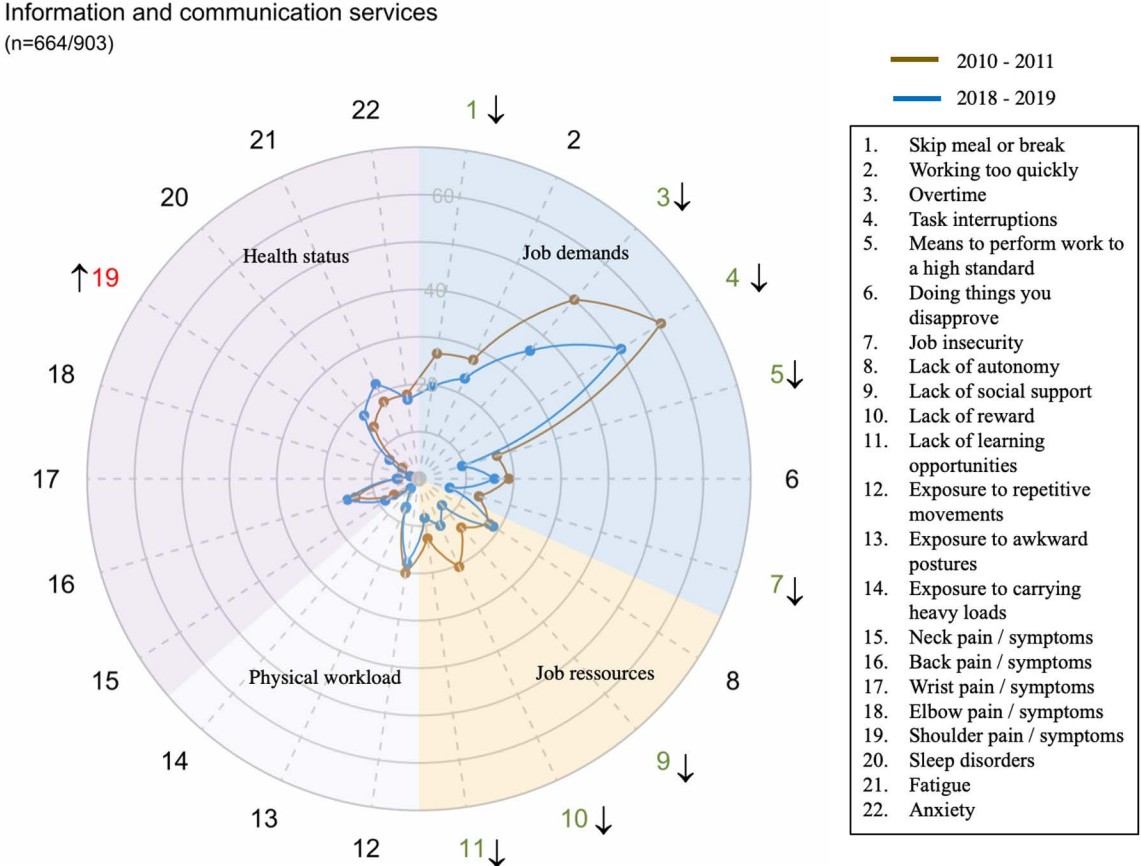

**Fig 8. Evolution of the percentage of workers exposed to psychosocial or physical risk factors and their health status in the information and communication services sector.**

Randomization of employees included in the study, on the basis of attendance to a periodic medical examination, and the use of sample weights for all the statistical analyses ensureed that our results can be extrapolated to the population of salaried workers in France, at least for workers belonging to the 13 sectors of activity studied. The random selection of the workers included in the study provided a more realistic view of their actual state of health and occupational exposure than the analysis of people referred to an occupational health center for pathological assessment, as is the case for the French National Network for Occupational Disease Vigilance and Prevention or the Danish Occupational Medicine Cohort[22,23].

Concerning the reliability of health status assessment, self-rated health status may be influenced by perceived occupational exposures [24], but our study did not only rely on self-assessment for health status since this part of the variable was assessed by an occupational health professional.

## Main evolutions

Globally, we observed improvement in workplace exposures and psychological and physical health status.

Job insecurity decreased in most sectors of activity over the period. This is consistent with the data of the French working conditions survey which observed no evolution of the prevalence of job insecurity between 2006 and 2013 [25,26].

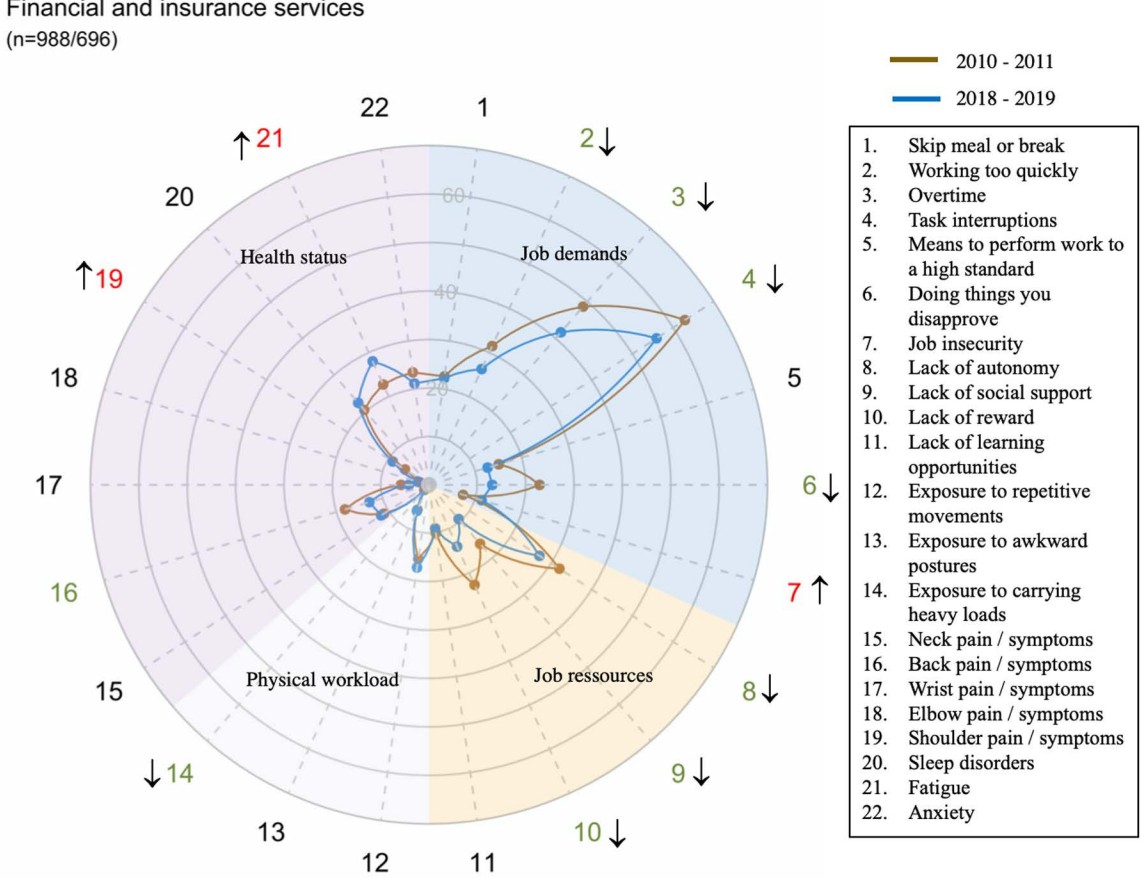

**Fig 9. Evolution of the percentage of workers exposed to psychosocial or physical risk factors and their health status in the financial and insurance services sector.**

Despite this global trend, job insecurity increased in two public sectors, *i.e.,* Electricity, gas, steam and air conditioning supply and Public administration and defense categories. This may seem surprising given the job security enjoyed by civil servants in France, but it probably reflects the fact that these civil servants are not sure whether their health status will allow them to keep on working in the forthcoming years [11].

The prevalence of musculoskeletal disorders decreased over the period, which is consistent with the decrease in exposure to psychosocial risk factors and physical workload since both are linked [20,27–29]. Other national or international surveys have also described a decreasing trend in work-related musculoskeletal disorders in the last decades, either globally [27,30–32], for specific localizations such as carpal tunnel syndrome [33] or in specific activities such as in the US construction industry [34]. Nevertheless, they was an increase in the Italian agricultural sector between 2004 and 2017 [35]. The limit of some of these studies is that they relied on compensation claims, which are notably under-reported [36], whereas the method used in our study was not impacted by such bias.

Sleep disorders were widely represented, affecting between 20% and 30% of the subjects participating in the study, and this proportion remained relatively constant, despite the improvement of working conditions. Yet, a French cross-sectional study has previously identified an association between psychological demands, low social support, low recognition, emotional demands and sleep disorders[37]. We had no information on shift work of employees in our study,

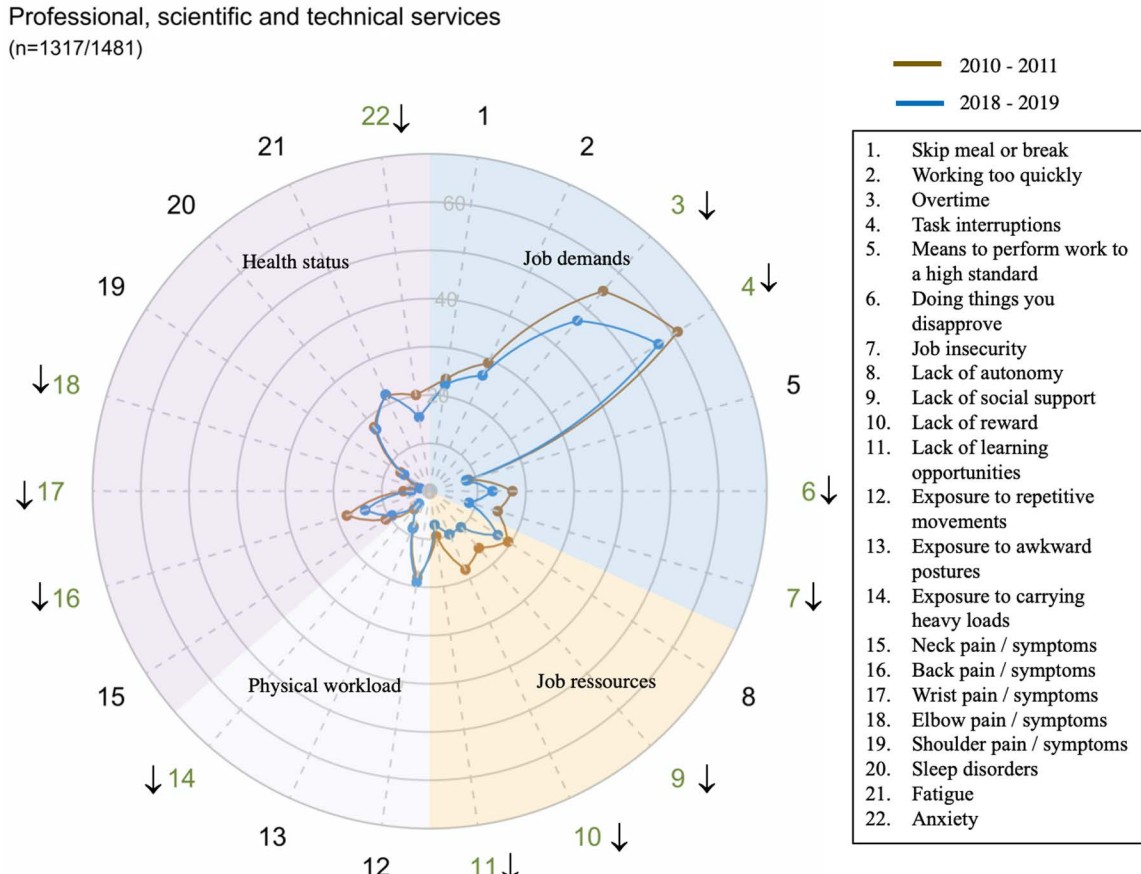

**Fig 10. Evolution of the percentage of workers exposed to psychosocial or physical risk factors and their health status in the professional, scientific and technical services sector.**

although it is a significant risk factor for sleep disorders. However, we observed an increase of sleep disorders in the Financial and insurances services sector, which is not exposed to shift-work.

The proportion of employees indicating that they had suffered from fatigue in the previous week increased in 85% of sectors between 2010–2011 and 2018–2019. Physical exposures are associated with increased bodily fatigue, the most marked associations being found when summing the number of different exposures [38]. However, the present study found no increase in exposure to physical factors which could provide a potential explanation for the observed increase in employees' fatigue. It could be hypothesized that the occurrence of fatigue is partly attributable to compassion fatigue, given its increased prevalence among sectors of activity characterized by frequent contact with the public in our study.

It is imperative that particular attention be given to a specific area of activity, owing to the unfavorable evolution of several parameters, *i.e.,*. Human health and social work activities. In the period spanning from 2010–2011–2018–2019, it was observed that seven out of eight health disorders exhibited a marked progression, with four of these disorders showing a significant escalation in severity. Concurrently, it was noted that the prevailing working conditions remained relatively constant, but to a high level, especially for working overtime and task interruptions. In the Swedish Longitudinal Occupational

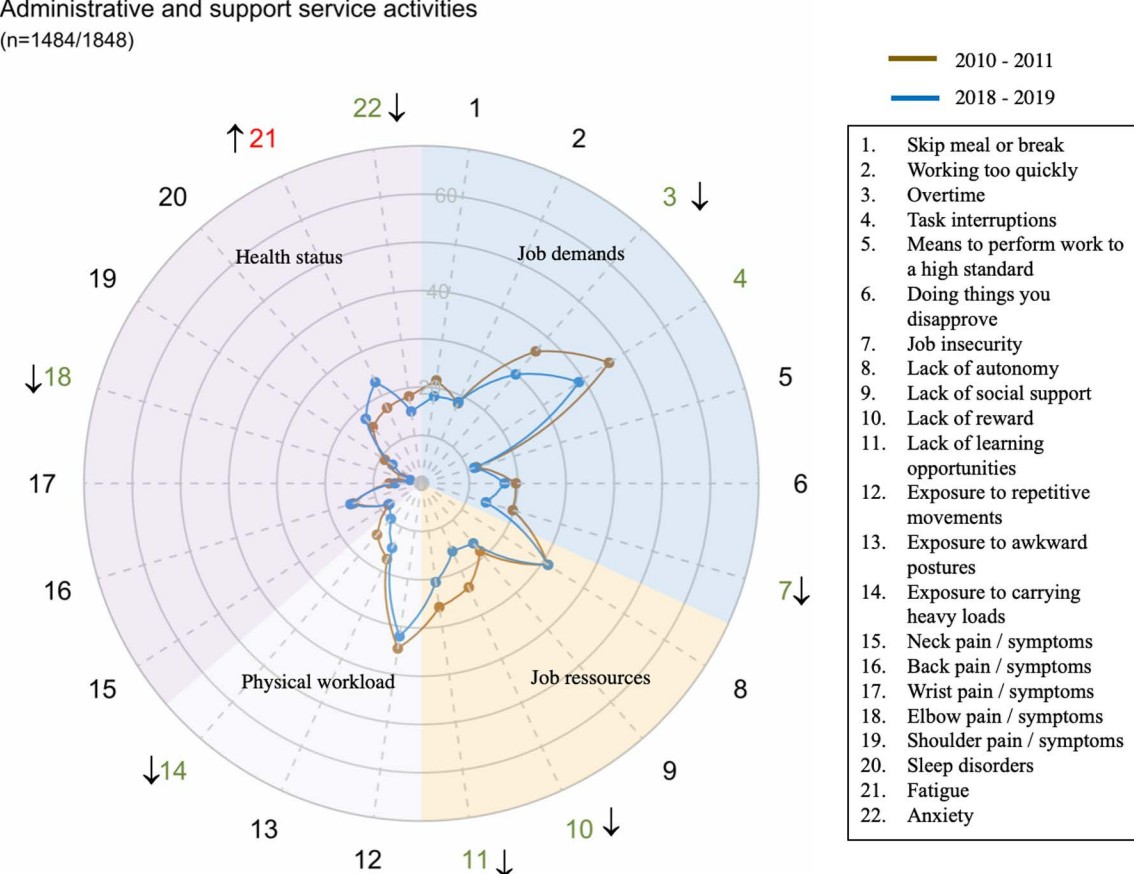

**Fig 11. Evolution of the percentage of workers exposed to psychosocial or physical risk factors and their health status in the administrative and support service activities sector.**

Survey of Health, workplace violence and low decision authority was considered to mediate the association between employment in the health and social care industry and sickness absence[39].

## Differences between sectors of activity

Beyond the overall trends, we observed major disparities between the different sectors of activity, both in terms of occupational exposure and the state of health of their employees.

Previous studies have shown that exposures to job stressors vary between working population groups in industrialized countries, which contributes to occupational health disparities [40,41]. Comparing trends in job demands, decision authority, and social support in 7 different categories of industry, Cerdas et al. observed different directions developments between 1991–2013 with high differences between industry categories [42].

What could explain the different trends among sectors of activity observed in our study?

First of all, we could think that occupational health policies or interventions specific to some activity sectors could explain why some sectors showed improvement while others did not. However, we are not aware of such specific interventions in the period studied. Moreover, regulations concerning occupational health in France are very detailed and strict and they apply to all employees. It is therefore unlikely that actions at a national level could have been implemented,

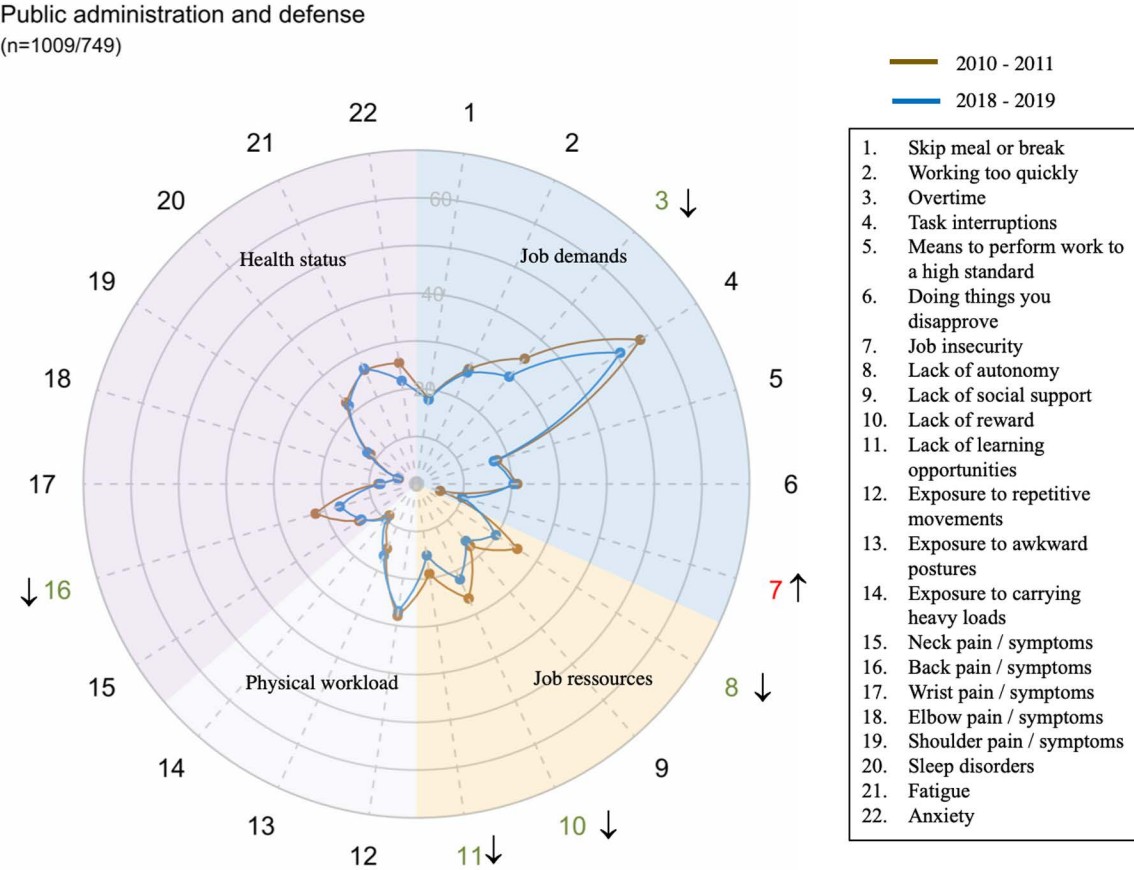

**Fig 12. Evolution of the percentage of workers exposed to psychosocial or physical risk factors and their health status in the public administration and defense sector.**

without us knowing it. Local actions may have occurred, but they cannot explain changes at a national level. We believe that the changes we observed are probably linked to economic changes.

Differences between sectors of activity may be linked to different compositions in terms of gender, age, employment arrangement, socioeconomic and social class positions or occupational skill level [4,5,40–42]. For example, in the Swedish work force, a negative trend of job demands and decision authority was observed in the female-dominated industries between 1991 and 2013, and, in 2020, the highest prevalence of problem drinking was in male-dominated Goods and Energy Production (7.7%), and the lowest was in female-dominated Health and Social Care (4.7%) [42,43]. However, some disparities between industry categories are independent of these factors. In France, the prevalence ratio of job insecurity is 1.4 for the private sector compared to the public one and blue-collar workers, particularly unskilled blue-collar and industrial workers, have a higher prevalence of job insecurity among both genders [26]. In our study, the apparent reduction in job insecurity could therefore be the result of a reduction in the proportion of employees and manual workers in the 2018–2019 cohort compared with the 2010–2011 cohort.

Human health and social work activities
(n=2428/3436)

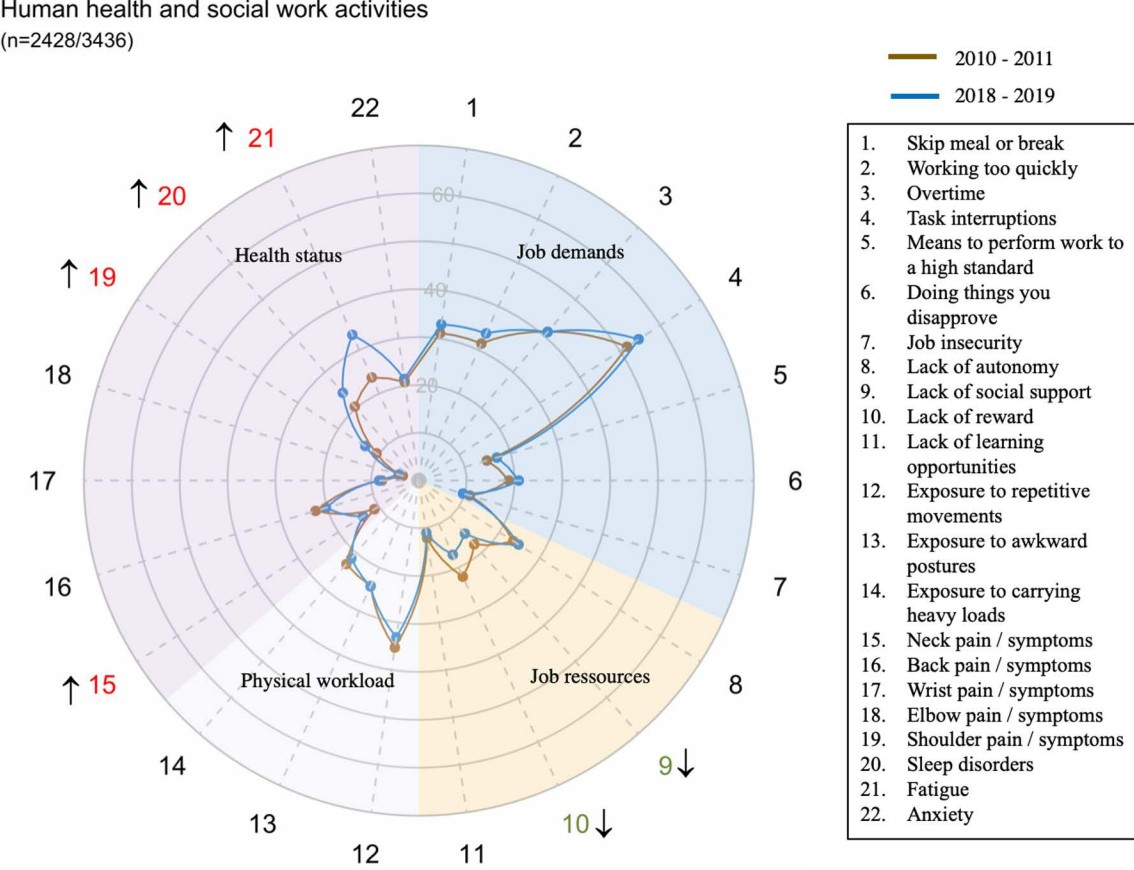

**Fig 13. Evolution of the percentage of workers exposed to psychosocial or physical risk factors and their health status in the human health and social work activities sector.**

Another possible determinant of such differences is an unequal distribution of company characteristics, such as the economic activity of the company and its implications for the volatility of the market for its products or services, differences in employment norms associated with different types of ownership, and differences in human resource policies related to organizational size [26,44].

## Conclusion

In general, the evolution and prevalence of occupational exposures or health problems do not provide equivalent information. It is possible to observe an improvement in an industry category over time, even if the prevalence of exposure or health complaint remains higher than in other categories where the prevalence has increased or remained stable.

It is imperative to comprehend the extent to which psychosocial and physical working conditions and health status vary across diverse industry categories, and to investigate the determinants of such disparities, in order to ensure that policy and practice interventions to reduce disparities can be appropriately directed.

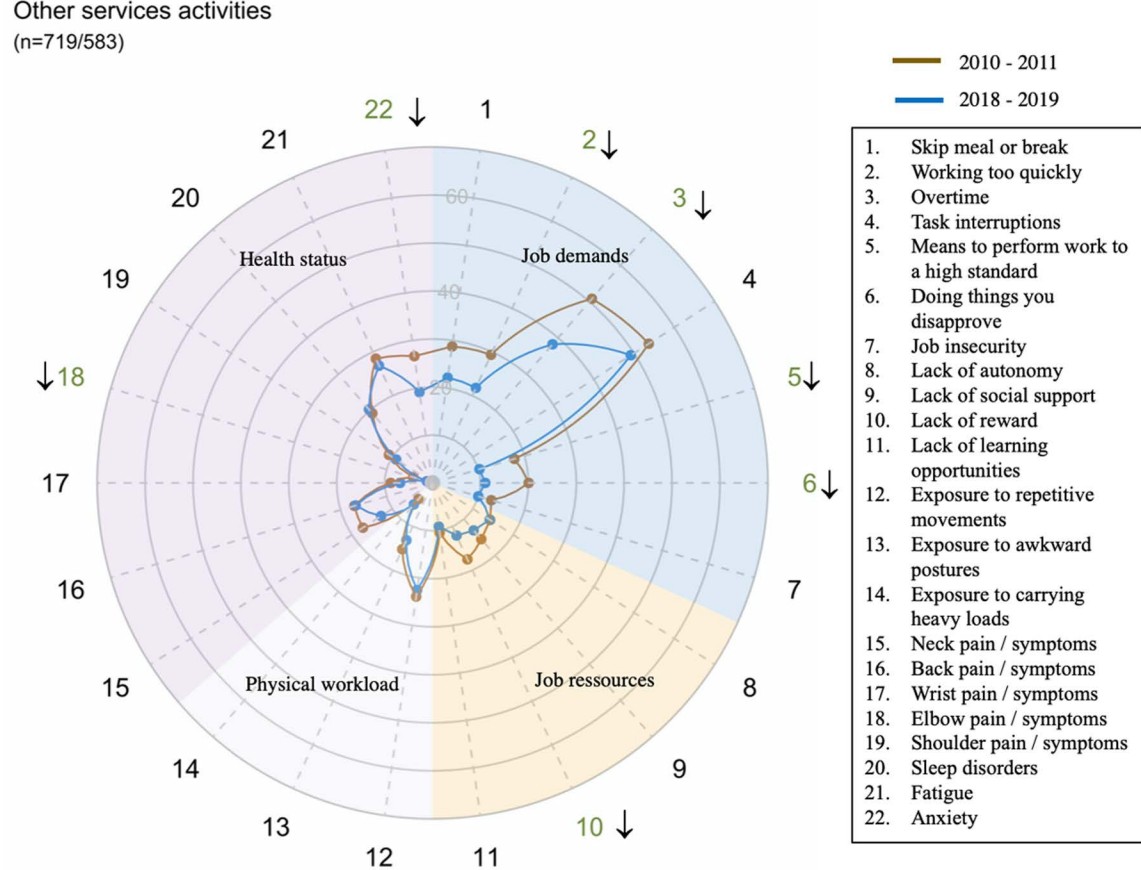

**Fig 14. Evolution of the percentage of workers exposed to psychosocial or physical risk factors and their health status in the other services activities sectors.**

From an interventionist perspective, the understanding of how work requirements and resources vary between sectors of activity, independent of the types of occupations, could facilitate the identification of sectoral factors in addition to general differences between occupations. Consequently, intervention policies in the domain of health and safety could be more effectively targeted towards specific sectors of activity. The health sector, for instance, appears to be experiencing an unfavorable trend, a situation that has likely been exacerbated by the pandemic.

Furthermore, the study of trends is of paramount importance for the identification of new risk factors, the determination of the success of occupational health interventions on a national level, and for the quantification of the health impact of various hazards [45].

Finally, it is imperative to maintain ongoing monitoring of the health of workers, as the impact of changes in working conditions may not be immediately apparent, particularly for conditions with a long latency period. Observatories such as EVREST, which rely on randomization of participants, should be promoted since they can provide relevant answers to those questions.

**Table 3. Changes of the proportion of workers concerned by each item from 2010-2011 to 2018-2019 for each sector of activity.**

| Sector of activity* | | C | D | F | G | H | I | J | K | M | N | O | Q | S | Global | Global p-value |
|---|---|---|---|---|---|---|---|---|---|---|---|---|---|---|---|---|
| **Working conditions** | | | | | | | | | | | | | | | | |
| Time pressure | 1-Shorten meal times | +6 | −21 | −4 | −2 | +5 | +3 | −26 | −2 | −5 | −15 | −1 | +6 | −23 | **−6** | 0.06 |
| | 2-Complete too quickly an operation | −23 | −31 | −4 | −13 | −3 | −15 | −16 | −17 | −10 | +2 | −3 | +7 | −26 | **−9** | 0.003 |
| Overtime | 3-Working overtime | +5 | −16 | +6 | −10 | −2 | +3 | −28 | −14 | −15 | −17 | −14 | −0 | −25 | **−6** | 0.007 |
| Task interruptions | 4-Need to stop his task | −10 | −16 | −12 | −3 | −1 | −7 | −16 | −11 | −8 | −16 | −9 | +6 | −8 | **−6** | 0.002 |
| Ethical dilemmas | 5-Insufficient means to perform his task | −24 | −44 | −44 | −32 | −3 | −38 | −45 | −16 | −6 | −4 | −4 | +15 | −43 | **−24** | 0.001 |
| | 6-Doing think he disapprove | −15 | −17 | −19 | −13 | +12 | −19 | −16 | −43 | −24 | −12 | −4 | +11 | −45 | **−16** | 0.001 |
| Job insecurity | 7-Fear of losing his job | −23 | +96 | −20 | −26 | −23 | −25 | −49 | +54 | −42 | −29 | +93 | −13 | −22 | **−11** | 0.11 |
| Autonomy | 8-Not choosing how to do | −4 | −20 | −23 | −11 | −6 | +19 | +6 | −15 | −13 | +0 | −21 | +6 | −1 | **−9** | 0.011 |
| Social support | 9-Lack of cooperation | −21 | −61 | −22 | −26 | −21 | −35 | −46 | −42 | −37 | −12 | −8 | −17 | −16 | **−23** | 0.001 |
| Reward | 10-Work not recognized | −35 | −38 | −34 | −26 | −36 | −44 | −47 | −38 | −45 | −35 | −17 | −23 | −31 | **−32** | 0.001 |
| Learning opportunities | 11-No opportunity to learn | −20 | −21 | −22 | −26 | −1 | −21 | −34 | −9 | −25 | −20 | −20 | −9 | −12 | **−20** | 0.001 |
| Physical workload | 12-Repetitive movements | −9 | +15 | −6 | −3 | −10 | −6 | −11 | +12 | +6 | −7 | −4 | −6 | −6 | **−5** | 0.06 |
| | 13-Awkward postures | −6 | −8 | −9 | +6 | −10 | −11 | −6 | +5 | +5 | −15 | +11 | +1 | −14 | **−5** | 0.18 |
| | 14-Carrying heavy loads | −0 | −32 | −2 | +5 | −15 | −10 | −10 | −51 | −31 | −31 | +12 | −7 | +34 | **−5** | 0.16 |
| **Health status** | | | | | | | | | | | | | | | | |
| Musculoskel-etal disorders | 15-Neck | −12 | −23 | −12 | −7 | −15 | −23 | +36 | +6 | −14 | −3 | −4 | +25 | −26 | −10 | 0.08 |
| | 16-Back | −8 | −26 | −26 | −7 | −3 | −8 | +12 | −29 | −22 | +4 | −24 | −10 | −2 | −15 | 0.001 |
| | 17-Wrist | −11 | −29 | +6 | −15 | +34 | +32 | +7 | −28 | −29 | −18 | −5 | +8 | −22 | −6 | 0.43 |
| | 18-Elbow | −8 | −51 | −0 | −8 | −23 | +30 | +12 | +25 | −35 | −39 | +6 | +26 | −70 | −12 | 0.21 |
| | 19-Shoulder | +6 | −30 | +15 | +4 | +24 | +3 | +75 | +51 | −13 | −21 | +7 | +27 | −17 | +10 | 0.14 |
| Psychological issues | 20-Sleep disorders | +4 | −8 | −9 | +5 | −7 | +25 | +21 | +10 | −4 | +14 | −4 | +19 | +6 | +4 | 0.34 |
| | 21-Fatigue | +3 | −16 | +15 | +12 | +6 | +3 | +23 | +23 | +1 | +34 | +1 | +41 | −6 | +11 | 0.007 |
| | 22-Anxiety | −29 | −31 | −47 | −15 | −6 | +2 | −6 | −10 | −23 | −17 | −15 | +3 | −28 | −20 | 0.001 |

The values in the boxes represent the percentage change between 2010–2011 and 2018–2019.

For example, the proportion of workers who declared being exposed to working overtime increased by 5% between 2010–2011 and 2018–2019 in the manufacturing sector but this change was not statistically significant, whereas the decrease of 28% in the Information and communication services sector was statistically significant.

| | |
|---|---|
| Statistically significant improvement | Statistically significant deterioration |

* C: Manufacturing; D: Electricity, gas, steam and air conditioning supply; F: Constructions and construction works; G: Wholesale and retail trade; repair of motor vehicles and motorcycles

H: Transportation and storage services; I: Accommodation and food service activities J: Information and communication services; K: Financial and insurance services; M: Professional, scientific and technical services; N: Administrative and support service activities; O: Public administration and defense; Q: Human health and social work activities; S: Other services activities

## Acknowledgments

The authors thank all EVREST national project team members.

## Author contributions

**Conceptualization:** Jean-Francois Gehanno, Laetitia Rollin.

**Data curation:** Jean-Francois Gehanno, Manon Couvreur, Laetitia Rollin.

**Formal analysis:** Jean-Francois Gehanno, Ariane Leroyer, Manon Couvreur, Laetitia Rollin.

**Investigation:** Serge Volkoff, Laetitia Rollin.

**Methodology:** Jean-Francois Gehanno, Ariane Leroyer, Manon Couvreur, Serge Volkoff, Laetitia Rollin.

**Project administration:** Jean-Francois Gehanno.

**Supervision:** Laetitia Rollin.

**Writing – original draft:** Jean-Francois Gehanno, Ariane Leroyer, Laetitia Rollin.

**Writing – review & editing:** Jean-Francois Gehanno, Ariane Leroyer, Serge Volkoff, Laetitia Rollin.

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
