## [Decision Letter · Decision Letter 0]

5 Feb 2025

Dear Dr. Gehanno,

Thank you for submitting your manuscript to PLOS ONE. After careful consideration, we feel that it has merit but does not fully meet PLOS ONE’s publication criteria as it currently stands. Therefore, we invite you to submit a revised version of the manuscript that addresses the points raised during the review process.

We look forward to receiving your revised manuscript.

Kind regards,

Swaantje Wiarda Casjens

Academic Editor

PLOS ONE

2. In the online submission form you indicate that your data is not available for proprietary reasons and have provided a contact point for accessing this data. Please note that your current contact point is a co-author on this manuscript. According to our Data Policy, the contact point must not be an author on the manuscript and must be an institutional contact, ideally not an individual. Please revise your data statement to a non-author institutional point of contact, such as a data access or ethics committee, and send this to us via return email. Please also include contact information for the third party organization, and please include the full citation of where the data can be found.

Additional Editor Comments:

Two reviewers have provided insightful feedback aimed at enhancing your manuscript. Both have raised concerns about the selection of sectors to be addressed, a point which I also found to be a critical issue. Furthermore, the terminology should be made more precise, as highlighted by the second reviewer, particularly in relation to terms like “sectors” and “professional categories.” In this context, I would also like to request that the terminology be reviewed and refined in the tables, with abbreviations consistently clarified in footnotes (e.g., sector abbreviations in Table 3). Additionally, please specify in Table 3 the nature of the “significant” improvement or deterioration—was this clinically/medically or statistically significant?

Other comments pertain to the data selection, particularly the time intervals and the focus on individuals born in October. Additionally, the presentation of statistical methods and the representation of results require further attention. Both reviewers also provided feedback on the discussion and conclusion sections. The inclusion of older references should be reconsidered, as I agree with Reviewer 2 that your results would benefit from comparison with more recent data.

Reviewers' comments:

Reviewer's Responses to Questions

**Comments to the Author**

1. Is the manuscript technically sound, and do the data support the conclusions?

Reviewer #1: Yes

Reviewer #2: Yes

2. Has the statistical analysis been performed appropriately and rigorously?

Reviewer #1: Yes

Reviewer #2: Yes

3. Have the authors made all data underlying the findings in their manuscript fully available?

Reviewer #1: Yes

Reviewer #2: No

4. Is the manuscript presented in an intelligible fashion and written in standard English?

Reviewer #1: Yes

Reviewer #2: Yes

Reviewer #1: Did any broader economic changes or trends in France during this period potentially influence the working conditions and health outcomes observed? How might these external factors have played a role?

Why were these 13 activity sectors chosen for analysis over the others defined by the National Institute of Statistics and Economic Studies? Was there a specific rationale for excluding the other sectors?

Given that only employees born in October were surveyed, how representative is the sample of the broader French workforce? Are there any potential biases introduced by this method?

Are there existing occupational health policies or interventions that could explain why some sectors showed improvement while others did not? Could you identify any sector-specific challenges that require policy changes or interventions?

Given that the data collection ended before the COVID-19 pandemic, it would be beneficial to include a brief discussion on how the pandemic might have influenced the observed trends, considering its significant impact on working conditions and health.

Some health outcomes, such as the increase in sleep disorders and fatigue, were highlighted but not deeply analyzed. Discuss potential reasons for these trends, such as changes in work environment, lifestyle factors, or stressors.

The conclusion could be enhanced by offering more concrete recommendations for policymakers or occupational health professionals on how to address the identified disparities across sectors.

Reviewer #2: This research article studies changes in physical exposure, psychosocial factors, psychological distress and musculoskeletal problems between the periods 2010-2011 and 2018-2019 across several sectors of activity. Based on the French Evrest database, it highlights an improvement in several physical and psychosocial constraints between the two periods, as well as a reduction in the percentage of musculoskeletal disorders and anxiety distributed across the different sectors. It also highlights the factors and diseases that have increased, as well as the differences between sectors.

The work carried out is of interest in occupational health by highlighting the factors requiring in-depth monitoring, as well as the sectors requiring the implementation of appropriate preventive measures. The choices made in the methodology should be clarified in order to provide a better understanding of the objective of this study and the work carried out.

I'm not qualified to assess the quality of English writing

General comments :

- Vocabulary: Starting from the abstract and in the rest of the paper, the vocabulary is very confusing. Lines 33 and 34 it is said “Job demand and job resources significantly improved […] physical workload decreased. Neck, elbow and back pain disorders significantly improved…”. The meaning of “decreased” and “improved” in this context is unclear. I suggest to remain more factual to enhance readability: something like “the proportion of workers reporting high job demand decreased”. At least define clearly what does “decrease” and “increase” mean in terms of proportions of exposures and diseases.

- Statistical unit of the study: The sector of activity is presented as the statistical unit. However, it is mentioned that it is also cross-referenced with professional categories (line 123). This leads to confusion as to what is being used. In addition, the nomenclatures used to code the NAF-2008 sector of activity and the PCS-ESE-2003 are also confusing, as the PCS-ESE-2003 is used to code occupations. Then in the tables and figures, the “professional category” is mentioned, but the labels seem to be activity sectors (NAF-2008 probably?). The statistical unit used should be clarified and standardised throughout the article.

- The choice of sectors of activity: Only 13 of the 21 sectors were studied, so it would be interesting to understand why this selection was made and what criteria were used.

- Presentation of Evrest database : the sections “the Evrest observatory” and “databases used” in the “Method” sections are not clearly defined, what is their respective purpose? The general information about the observatory are somewhat mixed with the subparts of data extracted and used in the study. I suggest to clarify, either with two clear sections or only one that gathers (1) all the information about Evrest and (2) the data management performed and its resulting sub-data. For example, I would expect the Table 1 in the “general description” of Evrest, whereas the list of main activity sectors explored should be in the description of the specific data extraction of this study.

- The choice of time periods: It would be interesting to know why these two periods were chosen and why the evolution of constraints was not studied over a continuous period. As far as I know, the Everest data is available for each year (since 2007?).

- The method for assessing the statistical significance of the evolution of results from 2010-2011 to 2018-2019 should be further explained. Even if the Rao-Scott test seem to be quite a classic, I’m not familiar with it. I understood it is more or less a Chi square test, but adjusted with the weights of observations. I’m not sure to really understand the relationship between the stratification variables of data used for the design of the weights, the objectives of the analysis (global statistics and sectoral statistics) and the adequacy of the test. In other words, I suggest to add one or two sentences about the choice of this test: why is this test really appropriate for this data? What are the assumptions (normality of data?) Maybe a literal description of the computation with an example should help. Note that this question is linked with the previous about the choice of the periods because the statistical significance of a trend over a continuous (or semi-continuous) period would be assessed with other methods.

- Base results : the provided data are focused on the differences between 2010-2011 and 2018-2019, but it would be interesting to discuss the level of exposures and the health status in 2010-2011. Indeed, the initial percentages of exposures and the percentages of workers suffering diseases are not clearly presented. As far as I understand, these values could be read on the figures and the levels of exposure are very different (around 50% for question 4 “task interruption” and almost zero for question 18 “elbow pain”). Please add a table or a figure at the beginning of the results.

- Improving the figures: The figures require further explanation to make them easier for readers to understand. I’m still unsure of what is presented (is it really the percentages of workers who answered “yes” to the 22 questions?). At least, a sentence reminding us of the link between the numbers in the figures and Table 1 would be useful. Better would be to use the labels instead of the numbering of questions. The legend mentions “significant improvement/degradation” when a cross is reported, but I can’t see any cross on the graph. It is difficult to read the set of graphs depicting the evolution for each activity sector. There are too much information. It is unclear whether the maximum is always 80%. The legends of these figures are missing.

- Structuring the results: The results may benefit from more structuration because the working conditions are presented first, then diseases and finally sectors of activity, but some of the information tends to overlap. A different presentation or the creation of sub-sections would perhaps make the reading easier.

- Improving the discussion 1: there is a lot of relevant information in the discussion but it is not really structured. It seem that the ideas are just presented as a list, adding transitions and linkage between ideas would help the reader.

- Improving the discussion 2: Lines 291 and following, two studies are largely used as comparison basis (Cerdas et al. and Swedish Work Environments Surveys), but these studies are old, and the world changed a lot since 1991. I’m not sure it is relevant to make any comparison between the period 1991-2013 of these studies and the period 2010-2019 explored in this paper. At least, reduce the part of the discussion dedicated to this comparisons or provide a better explanation of why you believe that your results are worth to be compared with such old data.

Specific comments

Authors : The symbol “&” is reported for authors who equally contributed to the work, but only Dr. Serge Volkoff has this sign associated with his name

Asbtract - Keywords: Add ‘physical’ to the term ‘occupational exposures’.

Introduction, Line 58 : define “macro analysis”

Introduction, Line 73: Explain what is meant by working conditions in this study (e.g. employee perceptions of psychosocial factors or physical exposures).

Introduction, Line 73: Specify the health problems studied in this article.

Method, lines 114-115 : a difference is mentioned about “public and private sectors” and “state public service”. The difference is unclear, in particular for non-French readers.

Method - The Evrest observatory, Line 123: Wouldn't it be (13*4) because 13 sectors of activity were selected? If not, then an explanation is missing.

Method - Databases used, Line 131 : 5 main axes concerning psychosocial constraints have been defined but are not subsequently used. Why not use them in the results?

Method, line 132 : Provide an English translation of “College d’expertise sur le suivi statistique des risques psychosociaux au travail »

Method - Ethics statement, Line 146 : Put a capital letter at the beginning of the sentence ‘Start and end ...’.

Result, Table 2: Why use the term “Professional categories”? Doesn't it correspond to the socio-professional categories shown in the last column of the table? Shouldn’t the term be changed into “Sector of activity”?

Result, Table 2: the numbers in the last two lines (global, at the very end of the table) are rigorously identical, which look like a copy/paste mistake.

Result, Line 203 and Line 207: Write the names of the sectors concerned.

Discussion, Line 287: Several sectors are mentioned, but only one is cited.

Discussion, line 320: You mention other studies about the decreasing trend in work related musculoskeletal disorders, but what about the trends in France? Why didn’t you mentioned the French health insurance data? As far as I know, the number of MSD compensated has greatly increased during the last decades (TMP n°57 RG at least)?

**Do you want your identity to be public for this peer review?** For information about this choice, including consent withdrawal, please see our Privacy Policy

Reviewer #1: No

Reviewer #2: No

---

## [Author Response · Author response to Decision Letter 1]

10 Apr 2025

Response to reviewers

PONE-D-24-25814

Changes in working and health conditions in different sectors between 2010 and 2019 in a representative sample of French Workers

PLOS ONE

Dear Dr. Gehanno,

Thank you for submitting your manuscript to PLOS ONE. After careful consideration, we feel that it has merit but does not fully meet PLOS ONE’s publication criteria as it currently stands. Therefore, we invite you to submit a revised version of the manuscript that addresses the points raised during the review process.

We look forward to receiving your revised manuscript.

Kind regards,

Swaantje Wiarda Casjens

Academic Editor

PLOS ONE

2. In the online submission form you indicate that your data is not available for proprietary reasons and have provided a contact point for accessing this data. Please note that your current contact point is a co-author on this manuscript. According to our Data Policy, the contact point must not be an author on the manuscript and must be an institutional contact, ideally not an individual. Please revise your data statement to a non-author institutional point of contact, such as a data access or ethics committee, and send this to us via return email. Please also include contact information for the third party organization, and please include the full citation of where the data can be found.

The contact point for accessing the data is the Data Protection Officer of the University of Rouen: dpo@univ-rouen.fr

We used a Data Availability Statement formulation inspired from the one of another study published in PLOS ONE (https://doi.org/10.1371/journal. pone.0287229):

Data Availability Statement:

The EVREST observatory was authorized by the Commission Nationale de l'Informatique et des Libertés (CNIL) on 4 March 2008. Following the General Data Protection Regulation (Regulation (EU) 2016/679), which came into force on 25 May 2018, this observatory was validated by the University of Rouen's Data Protection Officer and entered in the institution's data processing register under number 202009_008 and registered in the national HealthDataHub register (N° F20221129113417, https://www.health-data-hub.fr/projets/evolutions-et-relations-en-sante-au-travail-evrest). Data underlying the results are not publicly available due to legal reasons related to data protection and confidentiality. The data of the EVREST French observatory can be accessed on the basis of a research project submitted to the executive commitee. More information can be found on the following website: https://evrest.istnf.fr/_docs/Fichier/2017/4-170512013717.pdf

Additional Editor Comments:

Two reviewers have provided insightful feedback aimed at enhancing your manuscript. Both have raised concerns about the selection of sectors to be addressed, a point which I also found to be a critical issue. Furthermore, the terminology should be made more precise, as highlighted by the second reviewer, particularly in relation to terms like “sectors” and “professional categories.” In this context, I would also like to request that the terminology be reviewed and refined in the tables, with abbreviations consistently clarified in footnotes (e.g., sector abbreviations in Table 3). Additionally, please specify in Table 3 the nature of the “significant” improvement or deterioration—was this clinically/medically or statistically significant?

We agree and thank the Editor for this comment. Occupations and socio-professional categories (acronym, PCS) is a French statistical nomenclature used to classify occupations according to multiple factors, such as employment status (employee versus self-employed), the nature of the employer, the professional classification for employees, and the size of their company for the self-employed. Its name can be misleading and we were wrong using the term “professional categories” as a synonym of “Sector of activity”. We clarified and standardized this throughout the article

In Table 3, significant means statistically significant. It has been added in the text. Footnotes have been added.

Other comments pertain to the data selection, particularly the time intervals and the focus on individuals born in October. Additionally, the presentation of statistical methods and the representation of results require further attention. Both reviewers also provided feedback on the discussion and conclusion sections. The inclusion of older references should be reconsidered, as I agree with Reviewer 2 that your results would benefit from comparison with more recent data.

Response to reviewers

Reviewers' comments:

Reviewer's Responses to Questions

Comments to the Author

1. Is the manuscript technically sound, and do the data support the conclusions?

Reviewer #1: Yes

Reviewer #2: Yes

2. Has the statistical analysis been performed appropriately and rigorously?

Reviewer #1: Yes

Reviewer #2: Yes

3. Have the authors made all data underlying the findings in their manuscript fully available?

Reviewer #1: Yes

Reviewer #2: No

4. Is the manuscript presented in an intelligible fashion and written in standard English?

Reviewer #1: Yes

Reviewer #2: Yes

5. Review Comments to the Author

Reviewer #1: Did any broader economic changes or trends in France during this period potentially influence the working conditions and health outcomes observed? How might these external factors have played a role?

France has experienced four recessions since 1945: in 1975 (-0.9%), 1993 (-0.6%), 2009 (-2.9%) and 2020 (-7.5%). In 2008, the financial crisis of 2007 turned into an economic crisis and a global recession. In the United States, this began in early 2008. In European countries, and France in particular, it began at the end of 2008, following the worsening of the global financial crisis (bankruptcy of Lehman Brothers). Indeed, the 2009 recession in France was a direct consequence of the 2007-2008 financial crisis.

There was no crisis between 2009 (subprimes) and 2020 (COVID), although France came close to recession again in the first quarter of 2013, against the backdrop of the eurozone crisis. According to INSEE, all the engines of the French economy worked in slow motion: household consumption, although revised upwards, remained at half-mast, as did business investment, weighed down by weak margins. Foreign trade also suffered from an unexpected fall in German and British imports.

Why were these 13 activity sectors chosen for analysis over the others defined by the National Institute of Statistics and Economic Studies? Was there a specific rationale for excluding the other sectors?

We had not enough data for the 8 other sectors of activity, which employ a limited number of workers. We set the threshold at 500 questionnaires at least for one sector of activity and for both periods. For example, there were only 170, 47 and 20 workers included for “Arts, entertainment and recreation”, “Mining and quarrying” and “Activities of households as employers” sectors, respectively. The agricultural sector was also very poorly represented (only 77 workers included 2010), as it is subject to specific occupational health monitoring by a special social security system. It was therefore excluded from the survey (sector A ‘Agriculture, forestry and fishing’)

Given that only employees born in October were surveyed, how representative is the sample of the broader French workforce? Are there any potential biases introduced by this method?

We used the same methodology that the permanent demographic sample (échantillon démographique permanent – EDP) which was created in 1968 at INSEE (French National Institute for Statistics and Economic Studies) from the compilation of data from the population censuses and civil register. The aim was to set up a new tool for the analysis of geographical mobility and social trajectories, such as social differentials in mortality, professional trajectories over a long period or the trajectory of immigrants (professional mobility and acquisition of French nationality, for example). This EDP includes people born in October as randomization method.

The methods used to ensure representability of the sample have been described in another article (Ref 13): Weighting methodology for the national EVREST survey data

Objectives: The main objective was to describe the weighting methodology used for the national EVREST (Evolution and Relations in Health at Work) survey data. The secondary objectives were on the one hand to assess the extent of the differences between crude and weighted estimates, on the other hand to verify that the two-year gap in the availability of the reference data used does substantially not impact the estimates.

Methods: The study was based on data collected in 2013 and 2014 (N = 26,227). The weighting included 2 steps: 1) a first weighing to take into account the probability of participation of each employee; and 2) a calibration on margins to correct the potential distortions of the sample in comparison with the scope of the survey, the reference data used coming from the annual declarations of social data (DADS) of the years 2014 and 2012. The impact of the weighting method was studied using the differences between crude and weighted percentages for the 60 variables of the questionnaire.

Results: 90% of the differences between crude and weighted estimates were between – 2.0% and + 2.0% using the 2014 DADS, and 83% using the 2012 DADS. The most overestimated crude estimate concerned full-time work and the most underestimated was contact with the public. The impact of the two-year gap in the availability of the reference data used was weak.

Conclusion: A weighting methodology for EVREST survey was define and implement, allowing results to be extrapolated to the scope of the survey

Are there existing occupational health policies or interventions that could explain why some sectors showed improvement while others did not? Could you identify any sector-specific challenges that require policy changes or interventions?

We are not aware of such specific interventions in the period studied. However, regulations concerning occupational health in France are very strict and detailed and they apply to all employees. It is therefore unlikely that actions at a national level could have been implemented, without us knowing it. Local actions may have occurred, but they cannot explain changes at a national level. We believe that the changes we observed are probably linked to economic changes.

This was added in the manuscript.

Given that the data collection ended before the COVID-19 pandemic, it would be beneficial to include a brief discussion on how the pandemic might have influenced the observed trends, considering its significant impact on working conditions and health.

We probably did not understand well this comment but the COVID-19 pandemic cannot not have influenced the results for 2018-2019. However, we fully agree with the impact of this pandemic and we plan to compare 2018-2019 to 2023-2024 when the data will be consolidated.

We already mentioned in the manuscript that “A limitation of our study is that data were collected before COVID. The COVID-19 pandemic changed people's working conditions worldwide, induced income losses, increased work stressors and impaired the mental health of workers [7,15,16]. However, most studies on trends in working conditions and health are in the same situation, due mainly to the delay in collecting and analysing data at a national level. Furthermore, knowing the trends before the pandemic will be important to analyze the changes it induced.”

Some health outcomes, such as the increase in sleep disorders and fatigue, were highlighted but not deeply analyzed. Discuss potential reasons for these trends, such as changes in work environment, lifestyle factors, or stressors.

We agree and we developed these points in the discussion.

The conclusion could be enhanced by offering more concrete recommendations for policymakers or occupational health professionals on how to address the identified disparities across sectors.

We agree and we developed these points in the conclusion.

Reviewer #2: This research article studies changes in physical exposure, psychosocial factors, psychological distress and musculoskeletal problems between the periods 2010-2011 and 2018-2019 across several sectors of activity. Based on the French Evrest database, it highlights an improvement in several physical and psychosocial constraints between the two periods, as well as

---

## [Editor Report · Decision Letter 1]

15 Apr 2025

Dear Dr. Gehanno,

Thank you for submitting your manuscript to PLOS ONE. After careful consideration, we feel that it has merit but does not fully meet PLOS ONE’s publication criteria as it currently stands. Therefore, we invite you to submit a revised version of the manuscript that addresses the points raised during the review process.

We look forward to receiving your revised manuscript.

Kind regards,

Swaantje Wiarda Casjens

Academic Editor

PLOS ONE

Journal Requirements:

Additional Editor Comments:

Thank you for the revised manuscript. You have addressed the reviewer comments, and I believe these changes have significantly improved the manuscript. Nevertheless, I have a few minor comments that should still be addressed:

The explanation of Figure 1 is very well done. Please also specify what type of significance is being referred to here. Additionally, there is a typo: “2018-2018”. In general, statistically significant results should be represented differently than with the colors "red" and "green" in all figures due to the prevalence of congenital red-green color blindness.

How do Figure 1 and Figure 2 differ? If they are based on the same data, Figure 1, including the explanations, should be sufficient.

Further minor comments:

• Method, line 96: Provide an English translation of “College d’expertise sur le suivi statistique des risques psychosociaux au travail.

• Discussion, line 285: The term “limitation” should be used instead of “weaknesses.”

---

## [Author Response · Author response to Decision Letter 2]

17 Apr 2025

Additional Editor Comments:

Thank you for the revised manuscript. You have addressed the reviewer comments, and I believe these changes have significantly improved the manuscript.

We agree and we thank the reviewers for their valuable comments

Nevertheless, I have a few minor comments that should still be addressed:

The explanation of Figure 1 is very well done. Please also specify what type of significance is being referred to here.

We added “Statistically significant” in figure 1

In general, statistically significant results should be represented differently than with the colors "red" and "green" in all figures due to the prevalence of congenital red-green color blindness.

We have kept the colour in the figures but added a specific symbol for statistically significant deteriorations or improvements, to make them easier to read for people who cannot distinguish red from green.

Additionally, there is a typo: “2018-2018”.

This has been corrected

How do Figure 1 and Figure 2 differ? If they are based on the same data, Figure 1, including the explanations, should be sufficient.

We agree. We had displayed two figures since the first one was just to explain how to read the figures overall. We have removed Figure 2 and renumbered Figures 3 to Figures 2

Further minor comments:

• Method, line 96: Provide an English translation of “College d’expertise sur le suivi statistique des risques psychosociaux au travail.

We added the translation: “College of experts on the statistical monitoring of psychosocial risks at work”

• Discussion, line 285: The term “limitation” should be used instead of “weaknesses.”

It has been changed

---

## [Editor Report · Decision Letter 2]

23 Apr 2025

Changes in working and health conditions in different sectors between 2010 and 2019 in a representative sample of French Workers

PONE-D-24-25814R2

Dear Dr. Gehanno,

We’re pleased to inform you that your manuscript has been judged scientifically suitable for publication and will be formally accepted for publication once it meets all outstanding technical requirements.

Kind regards,

Swaantje Wiarda Casjens

Academic Editor

PLOS ONE
---

## [Editor Report · Acceptance letter]

PONE-D-24-25814R2

PLOS ONE

Dear Dr. Gehanno,

I'm pleased to inform you that your manuscript has been deemed suitable for publication in PLOS ONE. Congratulations! Your manuscript is now being handed over to our production team.

Kind regards,

on behalf of

Dr. Swaantje Wiarda Casjens

Academic Editor

PLOS ONE